**Soil trace gas fluxes along orthogonal precipitation and soil fertility gradients in tropical**
**lowland forests of Panama**
Amanda L. Matson[*1], Marife D. Corre[*1], Kerstin Langs[1] and Edzo Veldkamp[1]
[*]These authors contributed equally to this work
[1]Soil Science of Tropical and Subtropical Ecosystems, Büsgen Institute, University of Goettingen,
Buesgenweg 2, 37077, Goettingen, Germany
*Correspondence to:* Amanda L. Matson (amatson@gwdg.de)
**Abstract**
Tropical lowland forest soils are significant sources and sinks of trace gases. In order to model
soil trace gas flux for future climate scenarios, it is necessary to be able to predict changes in soil
trace gas fluxes along natural gradients of soil fertility and climatic characteristics. We quantified
trace gas fluxes in lowland forest soils at five locations in Panama, which encompassed
orthogonal precipitation and soil fertility gradients. Soil trace gas fluxes were measured monthly
for one (NO) or two ($CO_2$, $CH_4$, $N_2O$) years (2010-2012), using vented dynamic (for NO only) or
static chambers with permanent bases. Across the five sites, annual fluxes ranged from: 8.0 to
10.2 Mg $CO_2$-C ha$^{-1}$ yr$^{-1}$, -2.0 to -0.3 kg $CH_4$-C ha$^{-1}$ yr$^{-1}$, 0.4 to 1.3 kg $N_2O$-N ha$^{-1}$ yr$^{-1}$ and -0.82
to -0.03 kg NO-N ha$^{-1}$ yr$^{-1}$. Soil $CO_2$ emissions did not differ across sites, but did exhibit clear
seasonal differences and a parabolic pattern with soil moisture across sites. All sites were $CH_4$
sinks; within-site fluxes were largely controlled by soil moisture whereas fluxes across sites were
positively correlated with an integrated index of soil fertility. Soil $N_2O$ fluxes were low
throughout the measurement years, but highest emissions occurred at a mid-precipitation site
with high soil N availability. NO uptake in the soil occurred at all sites, with the highest uptake
at the low-precipitation site closest to Panama City; NO uptake was likely due to high ambient
NO concentrations from anthropogenic sources. Our study highlights the dual importance of
short-term (climatic) and long-term (soil/site characteristics) factors in predicting soil trace gas
fluxes.
*Keywords:* greenhouse gases, carbon dioxide, methane, nitric oxide, nitrous oxide, tropical forest

## 1 Introduction

Soils can be both sources and sinks of carbon dioxide ($CO_2$), methane ($CH_4$), nitrous oxide ($N_2O$) and nitric oxide (NO). Tropical forest soils, specifically, are the largest natural source of soil $CO_2$ (Raich and Schlesinger, 1992) and $N_2O$ (Bouwman et al., 1993; Prather et al., 1995) and can be significant sinks of $CH_4$ (Steudler et al., 1996; Keller et al., 2005; Sousa Neto et al., 2011). Although soil NO fluxes in tropical forests are often low (Keller and Reiners, 1994; Koehler et al., 2009b), and the canopy can act as a sink for a large proportion of soil-emitted NO (Rummel et al., 2002), even low emissions may be important in regulating atmospheric oxidant production (Keller et al., 1991; Chameides et al., 1992). However, annual soil trace gas fluxes in Central and South American (CSA) tropical lowland forests can vary significantly; in one study, $N_2O$ emissions varied by one order of magnitude (1.23 to 11.39 kg N ha$^{-1}$ yr$^{-1}$; Silver et al., 2005). Such disparity in measurements, caused by the temporal and spatial variability found in tropical forests (Townsend et al., 2008), makes it challenging to model soil trace gas fluxes from these areas and to predict how they might be affected by climate change.

Temporal variations in soil trace gas fluxes are primarily correlated with temperature and moisture. Temperature is often more important where there are annual extremes in temperature - such as in temperate and boreal regions - whereas precipitation and soil moisture are more important in tropical regions, where air temperature does not vary much throughout the year (Saikawa et al., 2013). Soil moisture affects microbial activity both directly through water availability and indirectly through its influence on the soil oxygen status and gas diffusivity (Davidson and Schimel, 1995). Spatial variations in soil trace gas fluxes are largely controlled by soil characteristics. Soil texture, for example, strongly influences soil water retention and gas diffusivity (Koehler et al. 2010; Hassler et al. 2015) as well as soil fertility, plant productivity,

decomposition and ultimately soil nutrient availability (Silver et al., 2000; Sotta et al., 2008;
Allen et al., 2015).

Soil $CO_2$ fluxes at the soil surface are the result of interacting belowground processes,

including autotrophic (root) respiration and heterotrophic (microbes and soil fauna) respiration
(Raich and Schlesinger, 1992; Hanson et al., 2000). Although temporal and spatial drivers may
be affecting these processes differently, the net response of soil $CO_2$ fluxes shows some
consistent trends. Soil $CO_2$ emissions from CSA tropical forest soils generally exhibit positive
relationships with soil temperature (Chambers et al., 2004; Schwendenmann and Veldkamp,
2006; Sotta et al., 2006, Koehler et al., 2009a) and soil moisture (Davidson et al., 2000). The
relationship between $CO_2$ and moisture is often parabolic, with emissions increasing until the
threshold at which anaerobic conditions start to inhibit soil $CO_2$ production and/or gas diffusion
and then decreasing (Schwendenmann et al., 2003; Sotta et al., 2006; Kohler et al., 2009a).
Spatial differences in soil $CO_2$ emissions can be affected by soil characteristics. Both Silver et al.
(2005) and Sotta et al. (2006) noted a soil texture effect on net soil $CO_2$ emissions; higher
emissions occurred in sandy as compared to clayey Ferralsol soils, which were attributed to
respiration from the higher fine root biomass in the sandy soils. Soil fertility can also affect net
soil $CO_2$ emissions; Schwendenmann et al. (2003) observed a positive relationship between soil
$CO_2$ flux and spatial differences in soil organic C and total N, and a negative relationship with
soil total P (possibly due to lower fine root biomass in areas of high P).

Soil $CH_4$ fluxes reflect the combined activity of both methanotrophs ($CH_4$ consumers)

and methanogens ($CH_4$ producers), the ratio of which can change in space and time. Since the
activity of both functional groups can increase with temperature (Conrad, 1996; Chin et al.,
1999; Mohanty et al., 2007), net changes of soil $CH_4$ fluxes in response to temperature are more
likely to be driven by other site conditions, such as soil moisture. Soil $CH_4$ fluxes (predominant
flux indicated by positive values (net emissions) or negative values (net consumption)) in CSA
tropical lowland forests often exhibit positive correlations with soil moisture (Keller and Reiners,
1994; Verchot et al., 2000; Davidson et al., 2004; Veldkamp et al., 2013) since high soil moisture
conditions favor $CH_4$ production, while $CH_4$ consumption is reduced due to inhibited diffusion of
$CH_4$ from the atmosphere to the soil (Le Mer and Roger, 2001; Koehler et al., 2012; Veldkamp et
al., 2013). Although they have less often been the focus of $CH_4$ studies, soil biochemical
characteristics (i.e. soil fertility status) may also play an important role. Veldkamp et al. (2013)
reported that increases in soil N availability stimulate $CH_4$ uptake and/or reduce $CH_4$ production
in soil, and Hassler et al. (2015) also showed that soil fertility (i.e. increased soil N availability
and decreased soil exchangeable Al) enhances soil $CH_4$ uptake.
N-oxide gases ($N_2O$ and NO) are produced and consumed through the microbial
processes of nitrification and denitrification (Chapuis-Lardy et al., 2007). In general, soil NO
production through nitrification dominates in aerobic conditions whereas soil $N_2O$ production
through denitrification dominates in anaerobic conditions (Conrad, 2002). Therefore, as shown in
several CSA tropical forest studies (Keller and Reiners, 1994; Verchot et al., 1999; Davidson et
al., 2004; Keller et al., 2005; Koehler et al., 2009b), with increases in soil moisture, soil NO
fluxes generally decrease (though Gut et al., 2002 show that this relationship is complex) while
soil $N_2O$ fluxes increase. Soil temperature can also be positively correlated with NO flux (Gut et
al., 2002), and negatively correlated with soil $N_2O$ emissions (Keller et al., 2005), though this
may be due to a co-correlation of soil temperature with soil moisture. Soil N-oxide fluxes may
also be affected by soil texture; soil $N_2O$ emissions can be stimulated by the higher soil N
availability and greater proportion of anaerobic microsites in clayey soils (Keller et al., 2005;
Silver et al., 2005; Sotta et al., 2008) whereas soil NO fluxes can be facilitated by the higher
diffusivity in sandy soils (Silver et al., 2005). Finally, as an essential substrate for nitrification
and denitrification, N availability in the soil is a primary controlling factor of soil N-oxide fluxes
(Koehler et al., 2009b; Corre et al., 2014).

Climate scenarios suggest that tropical regions may experience large changes in

precipitation regimes in the future, with moist tropical regions likely experiencing both higher
annual precipitation and more extreme precipitation events (Stocker et al., 2013). Such changes
could significantly alter current soil trace gas fluxes, since soil moisture – as described above –
plays an important role in both the temporal and spatial variability of soil trace gas fluxes. One
approach to studying how changes in precipitation may alter soil trace gas fluxes is to investigate
these fluxes along a natural gradient of climate (e.g. precipitation) in a localized region. This
approach was used by Holtgrieve et al. (2006) on the Kula volcanic series lava flow in Hawaii, to
show that soil N cycling and N-oxide fluxes were strongly affected by mean annual precipitation.
However, as suggested by Santiago et al. (2005), precipitation gradients in continental tropical
forests, where there are variations in species composition and soil parent material, may exhibit
different patterns than those from Hawaii. Additionally, precipitation (or climate) is itself a soil
forming factor (Jenny, 1945), and continental tropical lowland soils are considerably older than
the relatively young volcanic soils (i.e. Santiago et al., 2005). Therefore, soils of continental
precipitation gradients will reflect both the long-term effects of the precipitation regime (i.e. on
differences in soil physical and biochemical characteristics) in addition to short-term effects (i.e.
on soil moisture).

In this study, we quantified soil trace gas fluxes in tropical lowland forests of the Panama

Canal Watershed, spanning a precipitation gradient of 1700-3400 mm yr$^{-1}$ (Figure S1). Soil
fertility (based on an aggregate index that included clay content, [15]N natural abundance, effective
cation exchange capacity (ECEC), organic C:N ratio, and exchangeable Al; see 2.4) varied
orthogonally with this precipitation gradient (Figure S2). The objectives of our study were to: (1)
determine how soil fluxes of $CO_2$, $CH_4$, $N_2O$ and NO vary along orthogonal gradients of
precipitation and soil fertility, and (2) assess and compare the spatial and temporal controls of
soil trace gas fluxes in lowland tropical forests. By using orthogonal gradients of precipitation
and soil fertility, we were able to examine the relative importance of climatic factors vs. soil
biochemical characteristics for soil trace gas fluxes. We hypothesized that the temporal and
spatial patterns of soil trace gas fluxes across sites would follow the pattern of the most
important controlling soil factors: soil $CO_2$ fluxes would be parabolic in relation to increasing
soil moisture along the precipitation gradient; soil $CH_4$ fluxes would increase (or $CH_4$
consumption would decrease) with increasing soil moisture and decreasing soil fertility along the
precipitation gradient; and soil NO fluxes would decrease whereas soil $N_2O$ fluxes would
increase with increasing soil moisture along the precipitation gradient.

**2 Methods**
**2.1 Study sites**
Soil trace gas fluxes were measured in five study sites of the Center for Tropical Forest Science
(CTFS) located in the Panama Canal Watershed, central Panama (Table 1; Figure S1). Mean
annual air temperature is 27 °C (Windsor, 1990); the soil temperature across all sites fluctuated
between 22.5 and 27.5 °C during our study years (Fig. 1a). The five sites span a gradient of
annual precipitation  from 1700 mm yr$^{-1}$ in Metropolitan National Park (Met) on the Pacific side
to 3400 mm yr$^{-1}$ in P32 on the Atlantic side; the dry season generally lasts from January through
April (Corre et al., 2014). The sites were located in either old growth (P8 and P32) or mature
secondary (Met, P27, and P19) lowland forests, with tree densities (≥10 cm diameter at breast
height, DBH) of: 322 stems ha$^{-1}$ in Met, 395 stems ha$^{-1}$ in P27, 560 stems ha$^{-1}$ in P8, 520 stems
ha$^{-1}$ in P19, and 537 stems ha$^{-1}$ in P32 (Pyke et al., 2001). Since precipitation and parent
materials vary across these sites, soil  types also vary from Cambisols (Met and P27) on the
Pacific side to Ferralsols (P8, P19, and P32) on the Atlantic side (Table 1). Floristic composition
in these sites has been shown to be correlated with both regional precipitation and geology/soil
attributes (Pyke et al., 2001). The amounts and forms of soil organic P are strongly controlled by
soil properties whereas the proportion of soil organic P to total P is insensitive to the variation in
rainfall and soil properties (Turner and Engelbrecht, 2011).

**2.2 Soil trace gas flux calculation**
Soil $CO_2$, $CH_4$ and $N_2O$ fluxes were determined every 2-4 weeks from June 2010 through
February 2012 (28-31 sampling dates) using static vented chambers. Within each of the five
sites, a 20 m grid was placed over a 1 ha area and we randomly chose four 20 m x 20 m replicate
plots with a minimum distance of 20 m between plots. In each replicate plot, four permanent
chamber bases were installed (0.04 m$^2$ area and 0.25 m height after inserting 2 cm into the soil)
at the ends of two perpendicular 20 m transects that crossed in the plot's center. The total volume
of the chamber (with cover) was 11 L. To determine soil trace gas fluxes, chamber covers were
placed on the bases and gas samples (100 mL) were taken 2, 12, 22 and 32 min later. Samples
were stored in pre-evacuated glass containers with Teflon-coated stopcocks. At the Gamboa field
laboratory, gas samples were then analyzed for $CO_2$, $CH_4$ and $N_2O$ concentrations using a gas
chromatograph (Shimadzu GC-14B, Columbia, MD, USA) equipped with a flame ionization
detector (FID), an electron capture detector (ECD) and an autosampler, the same instrument that
was used in our earlier studies (Koehler et al. 2009a, 2009b, 2010, 2012; Veldkamp et al., 2013;
Corre et al. 2014). The instrument's detection limits were 50 ppm $CO_2$, 43 ppb $N_2O$ and 45 ppb
$CH_4$. Gas concentrations were measured by comparing integration peaks with those of three or
four standard gases containing increasing concentrations of $CO_2$, $CH_4$ and $N_2O$ (Deuste
Steininger GmbH, Mühlhausen, Germany).

Soil NO fluxes were determined every 2-4 weeks from June 2010 through June 2011 (18-

21 sampling dates) using open dynamic chambers (11 L volume) placed for 5-7 minutes on the
same permanent bases described above. The NO ambient mixing ratio was measured at a height
of 2 m above the ground (prior to each chamber measurement) near to each of the 4 chamber
locations at each of the 4 replicate plots per site on each sampling day. To measure NO, the air
from the chamber (ambient air) was sampled by a pump with a flow rate of 0.5-0.6 L $min^{-1}$, passed
through a $CrO_3$ catalyst that oxidizes NO to $NO_2$, and flowed across a fabric wick that is saturated
with a luminol solution. The luminol then oxidizes and produces chemiluminescence, which is
proportional to the concentration of $NO_2$, and is measured with a Scintrex LMA-3
chemiluminescence detector (ScintrexUnisearch, Ontario, Canada). To minimize deposition losses
within the sampling system, all parts in contact with the sample gas are made of Teflon (PTFE).
To prevent contamination of tubing and analyzers, particulate matter is removed from the sampled
air by PTFE particulate filters (pore size: 5 µm). In order to minimize potential changes in catalyst
efficiency caused by variations of air humidity, a known flux of ambient air dried by silica gel was
mixed to the sampled air to maintain a humidity of ~50 %; the detector was also calibrated in-situ
prior to and following chamber measurements, using a standard gas (3000 ppb NO;
DeusteSteininger GmbH, Mühlhausen, Germany). The instrument's detection limit was 0.04 ppb
NO/mV; mV is the electrical signal from the produced chemiluminescence.

Soil trace gas fluxes were calculated as the linear change in concentration over time, and

were adjusted for air temperature and atmospheric pressure measured during or directly after
sampling. To calculate soil NO fluxes, we considered the first 3 minutes of linear change in NO
concentrations with chamber closure time. For $CO_2$, $N_2O$ and $CH_4$ fluxes, all 3 gases were analyzed
in our gas chromatograph sequentially from the same gas sample. Thus, we based our best fit of gas
concentration vs. time on the $CO_2$ concentration increase, as it is the gas with the highest
concentration among these 3 gases. We did not observe any evidence of ebullition (e.g. sudden
increase of gas concentration during our 30-min chamber closure), and the $CO_2$ concentration always
increased linearly with time of chamber closure, so a linear fit was used for all 3 gases. Zero fluxes
and negative fluxes (i.e. for $N_2O$ and $CH_4$) were all included in our data analysis. Annual soil NO
fluxes were calculated using the June 2010-May 2011 measurements and annual soil $CO_2$ and
$N_2O$ fluxes were calculated using the January to December 2011 measurements; annual fluxes
were calculated using the trapezoid rule, assuming a linear relationship in fluxes between
sampling days (Koehler et al. 2009a, 2009b, 2010; Veldkamp et al., 2013; Corre et al. 2014).

**2.3 Soil biochemical characteristics**
In each replicate plot after each soil trace gas flux measurement, samples of the top 5 cm of soil
were taken about 1 m from each of the 4 chamber bases, pooled and mixed thoroughly in the
field to measure soil extractable $NH_4^+$ and $NO_3^-$ concentrations and gravimetric water content. In
the field, soil samples were placed into prepared extraction bottles containing 150 mL of 0.5M
$K_2SO_4$ and shaken thoroughly. Back at the field station ($\leq$ 6 h after samples were taken), the
extraction bottles were again shaken (~ 1 h) and then the extracts were filtered and frozen
immediately. The remaining soil was oven-dried at 105 °C for 1 day in order to ascertain
gravimetric water content; this was then used to calculate the dry mass of the soil that had been
extracted for mineral N. The frozen extracts were sent by air to the University of Göttingen,
Germany for analysis by continuous flow injection colorimetry (Cenco/Skalar Instruments,
Breda, Netherlands). The Berthelot reaction method was used to determine $NH_4^+$ (Skalar Method
155-000) and the copper-cadmium reduction method was used to determine $NO_3^-$ ($NH_4Cl$ buffer
without ethylenediaminetetraacetic acid; Skalar Method 461-000).

Soil pits were dug in the center of each of the four replicate plots per site and soil samples

were taken for the depth intervals of 0-5, 5-10, 10-25 and 25-50 cm. Soil samples were air-dried
and sieved through a 2-mm sieve. Natural abundance $^{15}N$ signatures were determined from the
ground soil samples using isotope ratio mass spectrometry (IRMS; Delta Plus, Finnigan MAT,
Bremen, Germany). We calculated the $\delta^{15}N$enrichment factor ($\epsilon$) using the Rayleigh equation
(Mariotti et al., 1981): $\epsilon = d_s - d_{so} / \ln f$, where $d_s$ is the $\delta^{15}N$ natural abundance at different depths
in the soil profile, $d_{so}$ is the $\delta^{15}N$ natural abundance of the reference depth (top 5 cm), and $f$ is the
fraction of total N remaining (i.e. the total N concentration at a given depth divided by the total
N concentration in the top 5 cm). The use of only surface $\delta^{15}N$ natural abundance values can be
limited, given its inherently high spatial variability (i.e. due to vegetation species differences and
surface topography). Therefore, we used not only the surface depth but also 4 depth increments
to determine the overall natural abundance enrichment factor ($\epsilon$). The $\epsilon$ value was used as an
integrative indicator of soil N availability, as this correlates with internal soil-N cycling rates
(Sotta et al., 2008; Baldos et al., 2015). Total organic C and N were measured from the ground
soil samples by dry combustion using a CN analyzer (ElementarVario EL; Elementar Analysis
Systems GmbH, Hanau, Germany). ECEC was determined from the sieved soil samples by
percolating with unbuffered 1M NH₄Cl and measuring the exchangeable element concentrations
(Ca, Mg, K, Mn, Na, Fe and Al) in the percolates using an inductively coupled plasma-atomic
emission spectrometer (ICP-AES; Spectroflame, Spectro Analytical Instruments, Kleve,
Germany). Base saturation was calculated as the ratio of exchangeable base cations to the ECEC.
Soil pH (H₂O) was analyzed from a 1:4 soil-to-water ratio. Particle size distribution of the
mineral soil was determined using the pipette method with pyrophosphate as a dispersing agent
(König and Fortmann, 1996).

**2.4. Soil fertility index**
The variation in soil types along our rainfall gradient (Table 1) was paralleled with variations in
soil biochemical characteristics (Table 2; see 3.1). Thus, we developed a soil fertility index using
principal component analysis (PCA), similar to the approach employed by Swaine (1996); for
each site, the index was based on five soil physical and biochemical properties: 1) clay content,
which reflects water- and nutrient-holding capacity, 2) ε that signifies long-term soil N status, 3)
ECEC and soil C:N ratio, which indicate bioavailability of rock-derived nutrients and soil
organic matter, and 4) exchangeable Al, which implies soil chemical suitability. We used the
depth-weighted average of these soil parameters (Table 2), measured at various depth intervals in
the top 50 cm depth (except for ε that is calculated for the whole depth; see above). The first
component factor of this PCA analysis explained 42 % of the variation in these soil
characteristics among sites (Figure S2) and the factor scores were used as the quantitative index
of soil fertility for each of the four replicate plots per site. This analysis showed that soil fertility
of the five lowland forests varied orthogonally with the precipitation gradient (Figure S2).

**2.4 Statistical analyses**
We note that our statistical tests are based on the four replicate plots in each of the five 1-ha
forest sites along these orthogonal gradients of precipitation and soil fertility, and that the sites
themselves were not replicated along the gradients. Consequently, our interpretations and
conclusions are limited only to these studied sites.
Soil trace gas fluxes (based on the average of the four chambers per replicate plot on each
sampling day) and the accompanying soil explanatory variables (soil temperature, gravimetric
moisture, $NH_4^+$ concentration and $NO_3^-$ concentration) were tested for normality using Shapiro-
Wilk's test; variables with non-normal distributions were square root or log transformed. We
then used linear mixed effects models (LMEs) to assess the differences in these repeatedly-
measured variables along the orthogonal precipitation and soil fertility gradients, with site and/or
season as the fixed effect(s) and sampling days and replicate plots as random effects. If the
Akaike information criterion (AIC) showed an improvement in the LME models, we included a
first-order temporal autoregressive function to account for the decreasing correlation of
measurements with increasing time (Zuur et al., 2009) and/or a variance function (varIdent) to
account for heteroscedasticity of fixed-factor variances (Crawley, 2012). To assess the
relationships between soil trace gas fluxes and soil explanatory variables, we used the mean
values of the four replicate plots on each sampling date, and conducted Pearson correlation tests
over the entire sampling period across the five sites and for each site. Lastly, we analyzed the
hierarchy of importance of the soil controlling factors of soil trace gas fluxes by selecting the
minimal adequate LME model. For this, we used a stepwise model simplification in which each
controlling factor was tested against a null model and the soil factor that showed the lowest AIC
value was ranked as the most important; the soil factors with the next lowest AIC values were
added step-wise into the model if this significantly improve the model fit. This analysis was
conducted on the mean values of the four replicate plots on each sampling date over the sampling
period across the five sites and for each site.

For the soil biochemical characteristics measured only once (Table 2), differences in

depth-weighted values (for the top 50 cm) among sites were evaluated using one-way analysis of
variance followed by a Tukey HSD test. Their relationships with soil trace gas fluxes across the
five sites (using annual values and average seasonal values) were tested using Spearman rank
correlations. In all statistical tests, differences among sites or between seasons, correlation
coefficients and minimal adequate LME models were considered significant at $P \leq 0.05$.
Data analyses were conducted using the R open source software (R Core Team, 2013).

**3 Results**
**3.1 Soil biochemical characteristics**
The soil $\delta^{15}N$ natural abundance signatures and $\varepsilon$, which are proxies of the long-term soil N
status (i.e. the higher the values, the higher the soil N availability), were lower at the low-rainfall
sites (Met and P27) than at one of the mid-rainfall sites (P19) ($P \leq 0.05$; Table 2). Soil organic C
was lower at one of the lower-rainfall sites (P27) than at the high-rainfall site (P32) whereas the
differences in total soil N among sites paralleled the increase in annual precipitation ($P \leq 0.05$;
Table 2). Soil pH, ECEC and exchangeable bases generally showed the opposite trend to that of
total soil N – higher values at the low-rainfall sites (with less-weathered soils) than at the mid-
and high-rainfall sites (with highly weathered soils) (all $P \leq 0.05$; Tables 1 and 2). Soil
exchangeable Al showed the converse pattern to that of exchangeable bases ($P \leq 0.02$; Table 2).
Of the four soil controlling factors that were monitored over time (temperature, moisture,
extractable $NH_4^+$ and extractable $NO_3^-$; Fig. 1a-d), only moisture and extractable $NO_3^-$ differed
strongly between seasons ($P < 0.01$; Fig. 1b-c; Table 3); soil moisture contents were higher in the
wet season than the dry season at all sites, while extractable soil $NO_3^-$ concentrations were lower
in the wet season that the dry season at all sites but P19. Temperature and extractable $NH_4^+$
exhibited between-season differences at only one site each (temperature - P8, extractable $NH_4^+$ -
P27; Table 3). Within each season, all four soil controlling factors differed along the
precipitation gradient (all $P < 0.01$ except $P = 0.04$ for extractable $NH_4^+$ in the wet season; Table
3). Soil temperatures in both seasons were lower at P32 (3400 mm) than at all other sites (not
significant at P27 in the dry season), and also lower at P27 (2030 mm) than Met (1700 mm). Soil
moisture contents, in contrast, were higher in both seasons at P32 than at the other four sites.
Extractable soil $NO_3^-$ concentrations in both seasons were higher at Met and P8 (2360 mm) than
at P27, P19 (2690 mm) and P32, and in the wet season, also higher at Met than P8. Extractable
soil $NH_4^+$ concentrations were higher at P32 than Met in both seasons. Across sites, over the 21-
month measurement period, soil moisture was inversely correlated with temperature (r = -0.28, $P$
$< 0.01$, n = 145) and extractable soil $NO_3^-$ (r = -0.51, $P < 0.01$, n = 145) and directly correlated
with extractable soil $NH_4^+$ (r = 0.46, $P < 0.01$, n = 145).

**3.2 $CO_2$ fluxes**
Although soil $CO_2$ emissions did not differ among the five sites over the 21-month measurement
period ($P = 0.40$; Fig. 2a; Table 3), emissions exhibited a parabolic relationship with soil
moisture across sites (Fig. 3) and were higher in the wet season than the dry season at each site
($P \leq 0.05$; Table 3). Over the 21-month sampling period, average daily soil $CO_2$ emissions from
the five sites were correlated with soil moisture (r = 0.35, $P < 0.01$, n = 145; Fig. 3), soil
temperature (r = 0.46, $P < 0.01$, n = 145), extractable soil $NH_4^+$ (r = 0.32, $P < 0.01$, n = 145) and
extractable soil $NO_3^-$ (r = -0.21, $P = 0.01$, n = 145); the dominant drivers in the wet season were
extractable $NH_4^+$ followed by temperature, while the dominant drivers in the dry season were
moisture, followed by temperature (Table S1). Within individual sites, daily soil $CO_2$ emissions
exhibited negative correlations with extractable soil $NO_3^-$ at Met (r = -0.48, $P = 0.01$, n = 27), P8
(r = -0.39, $P = 0.03$, n = 30), and P32 (r = -0.54, $P < 0.01$, n = 30). Moisture was a dominant
driver of $CO_2$ emissions from soils at all sites, with temperature (P27, P8 and P32) and mineral N
(Met, P19 and P32) both playing important roles as well (Table S1).

Similar to the relationship observed for average daily fluxes (Fig. 3), the annual soil $CO_2$

emissions (Table 4) also exhibited a parabolic pattern across the five sites of the precipitation
gradient: high at the mid-rainfall sites (P8 and P19) and low at both ends of the precipitation
gradient (Met and P32). There were no significant correlations between soil $CO_2$ emissions
(neither for annual $CO_2$ fluxes nor for wet- and dry-season averages) and the soil biochemical
characteristics (Table 5; Table S2).

**3.3 $CH_4$ fluxes**
On average, despite occasional emissions in the wet season (Fig. 2b), the soils in the five sites
acted as $CH_4$ sinks (Tables 3 and 4). Comparing between seasons, soil $CH_4$ uptake was higher in
the dry season than the wet season at all sites ($P \leq 0.05$; Table 3). Moisture was a dominant
driver of $CH_4$ flux in both seasons, but was stronger in the wet season (Table S1). Differences
among sites were the same in both seasons; soil $CH_4$ uptake at P19 (2690 mm) was higher than at
Met (1700 mm), P27 (2030 mm) and P32 (3400 mm), and higher at P8 (2360 mm) than at Met
($P \leq 0.05$; Table 3). Over the 21-month sampling period, average daily soil $CH_4$ fluxes from the
five sites were positively correlated (i.e. soil $CH_4$ uptake decreased) with soil moisture (r = 0.44,
$P < 0.01$, n = 145; Fig. 4a); moisture was also the dominant within-site driving factor at all sites
except Met (Table S1). Across sites, mineral N was a significant explanatory factor in both
seasons; within sites, this was only reflected in the model at P32 (Table S1) but average daily
soil $CH_4$ fluxes at P8 (r = -0.63, $P < 0.01$, n = 30), P19 (r = -0.48, $P < 0.01$, n = 28) and P32 (r =
-0.48, $P < 0.01$, n = 30) also exhibited negative correlations with extractable soil $NO_3^-$ (i.e. soil
$CH_4$ uptake increased as extractable soil $NO_3^-$ increased).
The annual soil $CH_4$ fluxes (Table 4) were positively correlated (Spearman *rho* = 0.84, *P*
< 0.01, n = 20; Fig. 4b) with the soil fertility index (Figure S2) and negatively correlated with
annual precipitation (*rho* = -0.63, $P < 0.01$, n = 20; Fig. 4c). Of the soil biochemical properties
measured once, annual soil $CH_4$ fluxes were negatively correlated with soil $^{15}N$ natural
abundance and exchangeable Al, and positively correlated with ECEC, base saturation and pH
(Table 5). Average seasonal soil $CH_4$ fluxes exhibited similar correlations (Table S2); it is
notable that when correlation analysis was separated by season, correlations with soil $^{15}N$ natural
abundance were stronger in the dry season than the wet season.

**3.4 $N_2O$ fluxes**
Soil $N_2O$ fluxes differed among sites only in the wet season and not in the dry season (Table 3;
Fig. 2c); soil $N_2O$ emissions in the wet season were higher at P8 (2360 mm) than all other sites
($P < 0.01$). Notably, the model fit also indicated no significant soil factors for the dry season, but
did identify $NO_3^-$ as a driving factor across sites in the wet season (Table S1). Within individual
sites, moisture was a controlling factor of $N_2O$ emissions at P8, P19 and P32, with $NO_3^-$
availability also important at P19 (Table S1). Comparing between sites, soil $N_2O$ emissions were
higher in the wet season than the dry season at P8 and P19 (2690 mm) ($P < 0.01$; Table 3). These
two sites were also the only two to exhibit correlations with soil controlling factors; soil $N_2O$
emissions increased with increases in soil moisture at P8 (r = 0.69, $P < 0.01$, n = 30) and P19 (r =
0.60, $P < 0.01$, n = 28), and decreased with increases in soil $NO_3^-$ concentration at P8 (r = -0.57,
$P < 0.01$, n = 30) and P19 (r = -0.38, $P = 0.05$, n = 28). Annual soil $N_2O$ emissions (Table 4)
were negatively correlated with clay content (Table 5). Seasonal average soil $N_2O$ emissions
were positively correlated with soil $^{15}N$ natural abundance in the wet season but not in the dry
season (Table S2).

**3.5 NO fluxes**
In all five sites, net uptake of NO was measured more often than net NO emissions from the soil
(Fig. 2d) and NO uptake was consistently higher ($P \leq 0.05$) in the wet than dry season, except at
P19 (2690 mm) where there was no difference between seasons (Table 3). Wet-season soil NO
uptake at Met (1700 mm) was larger than all other sites ($P < 0.01$; Table 3), while in the dry
season soil NO uptake at P19 was larger than at P8 (2360 mm) and P32 (3400 mm) ($P < 0.01$;
Table 3). Over the 13-month measurement period, there were no driving factors significant
across sites in the model fit (Table S1) but soil NO fluxes were negatively correlated (i.e. net NO
uptake increased) with ambient NO concentration ($r = -0.34$, $P < 0.01$, n = 103; Fig. 5). Within
individual sites, dominant drivers (Table S1) were moisture (P27 and P8) and temperature (P27),
with soil NO fluxes at P8 also exhibiting a negative correlation with soil moisture ($r = -0.67$, $P <$
0.01; n = 21) and positive correlation (i.e. net NO uptake decreased) with extractable soil $NO_3^-$ ($r$
= 0.65, $P < 0.01$; n = 21). There were no correlations with average seasonal soil NO fluxes in the
wet season, but in the dry season average seasonal soil NO fluxes were negatively correlated
with clay content across sites (Table S2).

**4 Discussion**
**4.1 $CO_2$ fluxes**
Soil $CO_2$ emissions from CSA tropical lowland forests, including Brazil (Davidson et al., 2000,
Chambers et al., 2004, Silver et al., 2005, Sotta et al., 2006), Puerto Rico (Raich and Schlesinger,
1992), Panama (Kursar 1989, Koehler et al., 2009a; Nottingham et al., 2010) and Costa Rica
(Schwendenmann and Veldkamp, 2006), range from 10.8 Mg C ha$^{-1}$ yr$^{-1}$ (Silver et al., 2005) to
39.7 Mg C ha$^{-1}$ yr$^{-1}$ (Sotta et al., 2006). Our annual soil $CO_2$ emissions (Table 4) were on the
lower end of this range. When compared with other studies in lowland forests of Panama, our
values were also at the lower end of those reported for Barro Colorado Island (BCI) (estimated at
14.5 Mg C ha$^{-1}$ yr$^{-1}$ in 1986; Kursar 1989) and Gigante (ranging from 13.59 ± 1.34 to 17.12 ±
1.59 Mg C ha$^{-1}$ yr$^{-1}$ between 2006 and 2008; Koehler et al., 2009a), which can, in part, be
attributed to inter-annual variation. Soil $CO_2$ fluxes at Gigante varied by more than 3 Mg C ha$^{-}$
$^1$yr$^{-1}$ between 2006 and 2008 (Koehler et al., 2009a), and fine litterfall, one of the substrates of
heterotrophic respiration, also varied by about 2 Mg ha$^{-1}$ yr$^{-1}$ from 1998 to 2008 (with annual
averages of 7.7-9.7 Mg ha$^{-1}$ yr$^{-1}$; Wright et al., 2011). Moreover, our values were comparable
with those of a mature secondary forest (P15 site, 7-18 Mg C ha$^{-1}$ yr$^{-1}$ in 2007/2008;
Notthingham et al., 2010) close to our P8 and P19 sites (Figure S1). Finally, three of our sites
(Met, P27 and P19) were mature secondary forests, with tree densities (particularly at Met and
P27; see 2.1) lower than the old growth forests on BCI (Pyke et al., 2001) and Gigante (Koehler
et al., 2009a). This may have additionally influenced soil $CO_2$ fluxes since up to 35 % of $CO_2$
emissions can be contributed by root respiration (Silver et al., 2005). Interestingly, regardless of
the contribution of autotrophic respiration to soil $CO_2$ fluxes, we did not detect any significant
differences in soil $CO_2$ fluxes among sites, but only found that across our 5 sites the temporal pattern
of soil $CO_2$ fluxes was strongly related to soil moisture.
Net soil $CO_2$ emissions responded to changes in climatic factors on a seasonal scale (i.e.
higher soil $CO_2$ fluxes in the wet than dry season at all sites; Table 3) and to daily fluctuations in
soil temperature and moisture across the five sites (see 3.2). The hierarchy of importance of the
soil factors are shown in Table S1: at each site (except P27) and during the dry season across
sites, soil moisture was the most important driving factor, followed by soil temperature, $NH_4^+$ or
$NO_3^-$, while during the wet season, when soil moisture was sufficient, the most important soil
factors were $NH_4^+$ and soil temperature (Table S1). The higher $CO_2$ emissions in the wet season
were likely due to the alleviation of water competition between decomposers and vegetation; in
seasonal tropical forests, litter tends to fall in the dry season, but low soil moisture limits
decomposition until the start of the wet season (Yavitt et al., 2004). Other studies from CSA
lowland forests have also reported a positive relationship between soil $CO_2$ emissions and soil
temperature (Chambers et al., 2004; Schwendenmann and Veldkamp, 2006; Sotta et al., 2006,
Koehler et al., 2009a), and parabolic relationships (Fig. 3) between soil $CO_2$ emissions and soil
moisture (Schwendenmann et al., 2003; Sotta et al., 2006; Kohler et al., 2009). Additionally, soil
$CO_2$ emissions responded to changes in soil mineral N both on the plot level and across sites (see
3.2). Relationships between soil $CO_2$ emissions and soil mineral N concentrations have not been
reported in other studies, although Schwendenmann et al. (2003) observed that spatial
differences in soil total N were positively correlated with soil $CO_2$ fluxes, and Koehler et al.
(2009a) found that chronic N addition decreased soil $CO_2$ fluxes in a montane tropical forest
(although not in a lowland forest). However, the correlations between $CO_2$ emissions and both
$NH_4^+$ (positive correlation) and $NO_3^-$ (negative correlation) may also simply be reflecting a co-
correlation between extractable mineral N and soil moisture (see 3.2).

In support of our hypothesis, we observed that annual soil $CO_2$ fluxes exhibited a

parabolic pattern along the precipitation gradient (Table 4) similar to the relationship seen with
the daily emissions and soil moisture (Fig. 3). However, as mentioned above, soil $CO_2$ efflux did
not differ among the five forest sites of this precipitation gradient (Table 3). This lack of
differences between sites could be due to similarity of a soil-controlling factor that results in
comparably low soil $CO_2$ emissions at all sites. For example, although organic C and total N
differed between sites, the soil C:N ratios were comparable along these orthogonal gradients of
annual precipitation and soil fertility (Table 2), suggesting that the bioavailability of soil organic
matter for heterotrophic respiration may be similar across sites. Additionally, the microbial
communities that contribute to heterotrophic respiration may have adapted to the existing
differences in substrate quantity (e.g. soil organic C), soil and climatic characteristics between
the sites (Tables 2 and 3) and therefore exhibited an overall similar soil $CO_2$ efflux.

**4.2 CH₄ fluxes**
Our findings show the scale-dependency of environmental controls on soil $CH_4$ fluxes – the
short-term (seasonal) pattern within and across sites were dominantly controlled by soil
moisture, temperature and mineral N (Table S1) whereas the long-term pattern based on annual
fluxes across sites was largely controlled by soil fertility (Fig. 4b).

The control of soil moisture on soil $CH_4$ fluxes has been shown in several CSA tropical

forest studies (Keller and Reiners, 1994; Verchot et al., 2000; Davidson et al., 2004; Veldkamp
et al., 2013). This was also observed at our sites, with less $CH_4$ uptake during periods of high
water content (i.e. wet vs. dry season; Table 3), soil moisture being the dominant controlling
factor at each site (except Met) and across sites during each season (Table S1), as well as a
positive correlation of soil $CH_4$ fluxes with water content (Fig. 4a). We attribute the dominant
role of soil moisture to controlling gas diffusivity from the atmosphere into the soil and/or
methanogenic activity during periods of high moisture. Our annual soil $CH_4$ uptake (Table 4)
was within the range of other reported values from Brazil and Panama (Verchot et al., 2000;
Davidson et al., 2004; Keller et al., 2005; Silver et al., 2005; Veldkamp et al., 2013). Studies that
have measured stronger uptake in CSA lowland forests (up to 4.90 kg C $ha^{-1}$ $yr^{-1}$; Keller and
Reiners, 1994; Steudler et al., 1996; Keller et al., 2005; Sousa Neto et al., 2011) may have had
soils with higher gas diffusivity due to lower soil water content and/or lower clay content (see
Veldkamp et al., 2013); in our five sites, the two sites with the highest sand content (P8 and P19;
Table 1) exhibited the highest soil $CH_4$ uptake (Tables 3 and 4). In addition to moisture, soil
$NO_3^-$ may also have been an important driver of temporal soil $CH_4$ uptake in our sites; we
observed increased $CH_4$ uptake as $NO_3^-$ concentrations increased in P8, P19 and P32 (see 3.3)
and it was a dominant controlling factor across sites in both seasons (Table S1). Although this
may have reflected a co-correlation between soil $NO_3^-$ concentration and soil moisture (see 3.1),
increasing $CH_4$ uptake in the soil with increasing mineral N has been observed in tropical forest
soils of Australia (Kiese et al., 2003), Panama (Veldkamp et al., 2013) and Indonesia (Hassler et
al., 2013). Additionally, our soils exhibited a correlation between annual soil $CH_4$ fluxes and soil
[15]N natural abundance signatures (Table 5), the latter being an indicator of soil N availability
(Sotta et al. 2008; Arnold et al. 2009; Baldos et al. 2015). When separated by season, the
correlation between soil $CH_4$ fluxes and soil [15]N natural abundance was stronger in the dry
season than the wet season (Table S2), supporting our claim that soil N availability enhanced
$CH_4$ uptake in soils when gas diffusion was favorable (dry season).

The control of soil fertility on the long-term pattern of soil $CH_4$ fluxes across sites was

depicted by a correlation between annual soil $CH_4$ fluxes and our calculated soil fertility index
(Fig. 4b), which exhibited an opposite pattern to that of annual precipitation (Figure S2). This
soil fertility control was supported by the strong correlations of both annual (Table 5) and
seasonal (Table S2) soil $CH_4$ fluxes with ECEC and exchangeable Al, both included in the soil
fertility index (Figure S2; see 2.4). The correlations between soil $CH_4$ fluxes and fertility
indicators reflected the site differences in soil biochemical characteristics (Table 2). Specifically,
as shown by the strong inverse correlation between soil $\delta^{15}N$ natural abundance signatures and
exchangeable cations (Table 5), the positive correlation between soil $CH_4$ flux and fertility (Fig.
4b) likely reflected the long-term effects of soil development (Tables 1 and 2) - more $CH_4$ uptake
occurred in highly weathered soils with less rock-derived nutrients but high soil N availability
(i.e. high $\delta^{15}N$ natural abundance signatures) (Tables 4 and 5). This supports our hypothesis that
soil $CH_4$ uptake reflected the control of soil moisture and N availability across sites along this
precipitation gradient. Our results also highlight the importance of considering soil properties - in
particular the degree of soil development - rather than simply climatic factors, when
predicting/modeling soil $CH_4$ fluxes on a large scale.

**4.3 $N_2O$ fluxes**
Our annual soil $N_2O$ fluxes (Table 4) were within the lower end of the range (1.23 - 11.4 kg N
$ha^{-1}$ $yr^{-1}$) reported from other CSA forest studies (Keller and Reiners 1994, Verchot et al., 1999,
Keller et al., 2005, Silver et al., 2005). In comparison with other studies from Panama, our $N_2O$
fluxes were similar to those measured from Gigante during dry years ($0.5 \pm 0.2$ kg N ha$^{-1}$ yr$^{-1}$ in
2008–2009 with annual precipitation 5–26 % lower than the 12-year average; Corre et al. 2014)
but slightly lower than those measured from the same site during wet years (1.0 - 1.4 kg N ha$^{-1}$
yr$^{-1}$ in 2006–2007 with annual precipitation 5–17 % higher than the 12-year average; Koehler et
al., 2009b). The low soil $N_2O$ fluxes at our sites were likely caused by the generally lower soil N
availability compared to the Gigante site; the five sites in our present study had an average gross
N mineralization rate of $4 \pm 1$ mg N kg$^{-1}$ d$^{-1}$ in the 2010 wet season (Corre et al. unpublished
data), which was significantly lower than those from Gigante ($29 \pm 6$ mg N kg$^{-1}$ d$^{-1}$ in the 2006
wet season; Corre et al. 2010).

Inter-annual variation in rainfall and hence soil moisture can also strongly affect soil $N_2O$

emissions (Corre et al., 2014). Our measured soil $N_2O$ emissions exhibited a tendency to be
higher in the wet season than the dry season (P8 and P19; Table 3), highest at the mid-rainfall
site of P8 (which could mean that at the high-rainfall sites $N_2O$ could have been further
denitrified to $N_2$), and were only correlated with the soil $^{15}N$ natural abundance signatures (as an
indicator of soil N availability) in the wet season (Table S2). At the sites (P8 and P19), where
$N_2O$ emissions were higher in the wet than dry season and soil $NO_3^-$ levels were lower in the wet
than dry season (Table 3), the inverse correlation between daily soil $N_2O$ emissions with $NO_3^-$
concentrations over the 21-month measurement period suggests that during the wet season $N_2O$
production could have been high but might have been further denitrified to $N_2$, and hence
resulted in low soil $NO_3^-$ concentrations. Although the reduction of $NO_3^-$ in the wet season could
also be caused by reduced nitrification, measurements in our study area (once in the wet and
once in the dry season) showed no significant differences between wet and dry seasons across
sites nor at each site (Corre et al. unpublished data). Additionally, gross nitrification was
correlated with $NO_3^-$ immobilization, but not with DNRA, suggesting that when there was high
$NO_3^-$ availability, this was preferably assimilated by the microbial biomass (Corre et al.
unpublished data). On the other hand, the soil $NO_3^-$ levels we show in Table 3 were measured
repeatedly, parallel to soil trace gas flux measurement, over our 21-month study period. The soil
$NO_3^-$ levels (Table 3) therefore reflected the concurrently occurring $NO_3^-$ production and
consumption processes. The argument that these reflect further denitrification to $N_2$ is supported
by our earlier study in Gigante, where nitrification and denitrification contributed equally to soil
$N_2O$ emissions during the dry season but denitrification was the main process contributing to soil
$N_2O$ emission in the wet season (Koehler et al., 2012; Corre et al. 2014). Our results partly
supported our initial hypothesis, in that soil $N_2O$ emissions were highest at the mid-precipitation
site (with the highest soil N availability as indicated by [15]N natural abundance; Table 2) due to
possible reduction of $N_2O$ to $N_2$ at the high precipitation site.

**4.4 NO fluxes**
Our annual soil NO uptake (Table 4) was considerably lower than other reported NO fluxes,
which are usually small net emissions rather than net uptake. Soil NO emissions from Panama,
Costa Rica and Brazil range from 0.26 to 7.88 kg N ha$^{-1}$yr$^{-1}$ (Keller and Reiners 1994, Verchot et
al., 1999, Gut et al., 2002, Keller et al., 2005, Silver et al., 2005, Koehler et al., 2009b; Corre et
al. 2014). However, the net uptake that we measured may be reflecting unusually high ambient
air NO concentrations in our forest sites as compared to forests from other studies. Although all
of our sites were located in mature-secondary or old-growth forests, the forests were located
within the Panama Canal watershed, where there is heavy, year-round marine traffic (~13,000
cargo ships in 2011; Hricko, 2012). Furthermore, the highest levels of soil NO uptake that we
measured were in the Met site (Table 4); in addition to being in the vicinity of the Panama Canal,
the park is located within the city limits of Panama City, which has a population of
approximately 1.6 million people (The World Factbook, 2015). Therefore, elevated ambient air
NO concentrations from anthropogenic emissions may be driving the NO uptake that we
measured. Our instrument cannot measure $O_3$ concentration, which could be high in these sites
influenced by anthropogenic emissions. Thus, the NO uptake that we saw may have been driven
by both chemical (Pape et al. 2009) and microbiological reactions (as NO is an intermediate
product of nitrification and denitrification; Davidson et al. 2000). The dominance of a chemical
reaction of NO uptake at our sites was supported by the fact that we observed a negative
correlation of soil NO fluxes with ambient air NO concentrations (i.e. net NO uptake increased
as ambient air NO concentration increased; Fig. 5). The reaction of NO with $O_3$, which is then
subsequently removed from the enclosed chamber air and deposited onto the soil, is driven by
the ambient air NO concentrations (Pape et al. 2009). This can occur in under a minute (which
we observed on days with low ambient air NO concentrations when we measured net soil NO
emissions; e.g. at P8 during the dry season, Fig. 2b) or can take up to the same order of
magnitude as the turnover time of the chamber air (which we observed on days with high
ambient air NO concentrations when we measured net NO uptake; e.g. at the Met site on most of
the sampling days, Fig. 2b). It is notable, that an earlier study in Gigante, which is also part of
the Panama Canal watershed, did not show net NO uptake but instead small net NO emissions
(Koehler et al., 2009b; Corre et al. 2014). However, as mentioned above, the Gigante site had
higher soil N-cycling rates (Corre et al. 2010) and lower ambient air NO concentrations than our
sites, such that NO production in the soil overrides the chemical reaction of NO uptake and thus
resulted in net soil NO emissions.
The general trend across sites did not support our hypothesis regarding soil NO emission,
since local conditions of high ambient NO concentrations in the atmosphere had an overriding
effect resulting in net NO uptake in soils (Fig. 2d). However, our results indicated that our soils
could also be a net source of NO when soil conditions were favourable and/or ambient air NO
concentrations were not elevated. We observed that net NO uptake was consistently higher in the
wet season than the dry season (Table 3); in the dry season, when aerobic soil conditions
prevailed due to low soil moisture contents (Table 3), NO production in the soil may have been
more favoured (Conrad, 2002), partly counteracting the chemical reaction of NO removal from
the atmosphere and its deposition onto the soil. This is also supported by the negative correlation
between dry-season soil NO fluxes and clay contents of the sites (Table S2), suggesting that soil
NO fluxes were responding to conditions favourable for NO production. Favourable soil
conditions were most visible at P8, which had the highest soil NO emissions (with low ambient
air NO concentrations) in the dry season (Table 3; Fig. 2d); soil NO fluxes at this site increased
when aerobic soil conditions prevailed (i.e. negative correlation with soil moisture; see 3.5) and
increased with substrate availability (i.e. positive correlation with soil $NO_3^-$; see 3.5).
In summary, although the soils in our study sites can be a net source of NO, particularly
during the dry season (Fig. 2d) and in sites where ambient air NO concentrations were low (Fig.
5), most of the time the soils acted as net sink of NO, signifying the importance of soil and
vegetation as NO sinks (Jacob and Bakwin, 1991; Sparks et al., 2001) in areas affected by
anthropogenic NO sources.

**4.5 Implications for climate change**

It is notable that, although all four trace gases were strongly correlated with the temporal variation in soil moisture and had clear differences between seasons (Table 3), there were no correlations between the soil trace gases when looking at the annual fluxes (Table 5) or seasonal averages (Table S2). This lack of correlation is presumably rooted in the interaction of other soil and/or climatic factors with known drivers of soil trace gas production and consumption; one future direction could be to do an in-depth analysis of the abundance/activity of functional microbial groups along these gradients of precipitation and fertility

We have shown that in the short term, soil trace gas fluxes were largely controlled by soil moisture, with the additional influences of soil temperature and mineral N concentration. However, in the long term and/or over large spatial scales, the degree of soil development and related soil fertility had a strong influence. Additionally, we have shown that even in presently undisturbed forests, gas fluxes can be affected by 'upstream' anthropogenic activities. Therefore, in order to understand and be able to predict soil trace gas fluxes under future climate scenarios, research needs to focus on identifying and predicting interacting effects of soil and site, as well as climatic characteristics, on soil-atmosphere trace gas exchange.

**Acknowledgements**

Funding for this study was provided by the Deutsche Forschungsgemeinschaft (DFG, Co 749/1-1) and by the Robert Bosch Foundation (Germany) for M.D. Corre's independent research group, NITROF. We gratefully acknowledge Dr. Helene Muller-Landau for hosting us and facilitating access to the field sites. The Smithsonian Tropical Research Institute and ANAM, Panama provided invaluable administrative and technical support. The efforts of the NITROF assistants

(Rodolfo Rojas and Erick Diaz), and the SSTSE laboratory technicians in completing the data
collection and analyses were much appreciated.

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

**Table 1** Description of location, rainfall and geology of one hectare forest inventory plots located in the Panama Canal watershed,
central Panama.

| Plot code [a] | Longitude, latitude | Elevation (m above sea level) [a] | Forest age classification [a] | Soil taxonomic order [b] | Soil texture (% sand/ silt/clay) [c] | Precipitation (mm yr$^{-1}$) [b] | Geology [b] |
|---|---|---|---|---|---|---|---|
| Metropolitan | 79º 33' W, 8º 59' N | 30 | mature secondary | Inceptisol (Cambisol) | 3/35/62 | 1700 | Aglomerate of andesitic tuff, Early-Late Oligocene |
| P27 | 79º 38' W, 9º 4' N | 160 | mature secondary | Inceptisol (Cambisol) | 2/38/60 | 2030 | Aglomerate of siltstone, tuff and limestone, Early Miocene |
| P8 | 79º 44' W, 9º 10' N | 50 | old growth | Oxisol (Ferralsol) | 12/39/48 | 2360 | Basaltic and andesitic lavas and tuff, pre-Tertiary |
| P19 | 79º 46' W, 9º 11' N | 160 | mature secondary | Oxisol (Ferralsol) | 10/27/63 | 2690 | Basaltic and andesitic lavas and tuff, pre-Tertiary |
| P32 | 79º 43' W, 9º 21' N | 340 | old growth | Oxisol (Ferralsol) | 1/39/60 | 3400 | Basaltic and andesitic lavas and tuff, pre-Tertiary |

[a] Plot codes and forest age classification are from Pyke et al. (2001).
[b] Turner and Engelbrecht (2011) reported the tentative soil order (based on US Soil Taxonomy with equivalent FAO classification in
brackets), mean annual precipitation (estimated from location and elevation data as described by Engelbrecht et al. 2007), and the
geological information (taken from Stewart et al. 1980).
[c] Textural analyses are the weighted average of the sampling depth intervals: 0-5, 5-10, 10-25 and 25-50 cm.

**Table 2** Soil biochemical characteristics in the top 50 cm of lowland forest soils along orthogonal gradients of annual precipitation (shown in brackets below each site) and soil fertility in the Panama Canal watershed, central Panama.

| Soil characteristics [a] | Metropolitan (1700 mm) | P27 (2030 mm) | P8 (2360 mm) | P19 (2690 mm) | P32 (3400 mm) |
|---|---|---|---|---|---|
| $\delta^{15}$N enrichment factor, $\varepsilon$ [b] | -1.95 ± 0.52 [b] | -0.37 ± 1.69 [b] | -2.76 ± 0.54 [ab] | -4.70 ± 0.44 [a] | -2.65 ± 0.30 [ab] |
| $\delta^{15}$N natural abundance (‰) | 5.9 ± 0.8 [c] | 6.3 ± 0.4 [bc] | 12.0 ± 1.0 [a] | 9.2 ± 0.9 [a] | 7.0 ± 0.3 [b] |
| Organic C (mg C g$^{-1}$) | 12.8 ± 1.7 [ab] | 10.8 ± 3.3 [b] | 15.1 ± 0.2 [ab] | 15.0 ± 1.3 [ab] | 19.6 ± 2.1 [a] |
| Total N (mg C g$^{-1}$) | 1.08 ± 0.15 [b] | 1.05 ± 0.25 [b] | 1.49 ± 0.02 [ab] | 1.44 ± 0.11 [ab] | 1.85 ± 0.17 [a] |
| C:N ratio | 10.9 ± 4.1 [a] | 9.07 ± 1.8 [a] | 9.76 ± 1.0 [a] | 9.88 ± 1.0 [a] | 10.1 ± 1.2 [a] |
| pH (1:4 H$_2$O) | 6.20 ± 0.46 [a] | 5.82 ± 0.72 [a] | 5.05 ± 0.17 [b] | 4.88 ± 0.30 [b] | 5.14 ± 0.22 [b] |
| ECEC [c] (mmol$_c$ kg$^{-1}$) | 199 ± 72 [ab] | 267 ± 11 [a] | 56 ± 2 [c] | 51 ± 6 [c] | 118 ± 12 [bc] |
| Exch. bases [c] (mmol$_c$ kg$^{-1}$) | 198 ± 72 [a] | 264 ± 10 [a] | 37 ± 6 [c] | 21 ± 8 [c] | 90 ± 11 [b] |
| Exchangeable Al (mmol$_c$ kg$^{-1}$) | 0.22 ± 0.13 [b] | 1.96 ± 0.51 [b] | 12.2 ± 4.7 [ab] | 22.6 ± 7.3 [a] | 22.2 ± 3.2 [a] |

[a] Means (±SE, $n$ = 4) followed by different letters indicate significant differences between sites (one-way ANOVA with Tukey HSD at $P \leq 0.05$). Values for each replicate plot are weighted average of the sampling depth intervals of 0-5, 5-10, 10-25 and 25-50 cm.

[b] Calculated using Rayleigh equation (Mariotti et al. 1981): $\varepsilon = d_s - d_{so} / \ln f$; $d_s$- $\delta^{15}$N natural abundance signatures at various depths in the soil profile, $d_{so}$- $\delta^{15}$N natural abundance of the reference depth (top 5cm) and $f$ is the remaining fraction of total N (i.e. total N concentration at a given depth divided by the total N concentration in the top 5 cm).

[c] ECEC – Effective cation exchange capacity; Exch. bases – sum of exchangeable Ca, Mg, K, Na

**Table 3** Soil factors (measured in the top 5 cm of soil) and trace gas fluxes from lowland forest soils along orthogonal gradients of
annual precipitation (mm per year; shown in brackets below each site) and soil fertility in the Panama Canal watershed, central
Panama.

| Site / season [a] | Soil temperature ($^\circ$ C) | Soil moisture (g g$^{-1}$) | Soil $NH_4^+$ (mg N kg$^{-1}$) | Soil $NO_3^-$ (mg N kg$^{-1}$) | $CO_2$ flux (mg C m$^{-2}$ h$^{-1}$) | $CH_4$ flux ($\mu$g C m$^{-2}$ h$^{-1}$) | $N_2O$ flux ($\mu$g N m$^{-2}$ h$^{-1}$) | NO flux ($\mu$g N m$^{-2}$ h$^{-1}$) |
|---|---|---|---|---|---|---|---|---|
| *Wet season* | | | | | | | | |
| Metropolitan (1700) | 25.8 (0.4)$^a$ | 0.64 (0.04)$^{Ac}$ | 5.94 (1.52)$^b$ | 1.95 (0.71)$^{Ba}$ | 126 (26)$^A$ | 1.47 (3.66)$^{Aa}$ | 5.78 (2.69)$^b$ | -11.6 (7.08)$^{Bb}$ |
| P27 (2030) | 25.2 (0.4)$^b$ | 0.72 (0.06)$^{Ab}$ | 6.39 (1.35)$^{Aab}$ | 0.51 (0.17)$^{Bc}$ | 124 (18)$^A$ | -3.01 (4.20)$^{Aa}$ | 4.15 (2.56)$^b$ | -3.24 (2.68)$^{Ba}$ |
| P8 (2360) | 25.6 (0.4)$^{Aab}$ | 0.60 (0.03)$^{Ac}$ | 5.68 (0.94)$^{ab}$ | 1.32 (0.54)$^{Bb}$ | 131 (19)$^A$ | -7.87 (6.95)$^{Abc}$ | 13.5 (7.0)$^{Aa}$ | -3.95 (6.60)$^{Ba}$ |
| P19 (2690) | 25.5 (0.5)$^{ab}$ | 0.72 (0.06)$^{Ab}$ | 7.29 (1.39)$^{ab}$ | 0.46 (0.39)$^c$ | 129 (15)$^A$ | -13.0 (6.92)$^{Ac}$ | 5.58 (3.13)$^{Ab}$ | -3.98 (4.95)$^a$ |
| P32 (3400) | 24.6 (0.4)$^c$ | 0.90 (0.08)$^{Aa}$ | 8.21 (1.87)$^{Aa}$ | 0.49 (0.27)$^{Bc}$ | 107 (17)$^A$ | -6.79 (6.09)$^{Aab}$ | 6.41 (3.09)$^b$ | -4.01 (4.34)$^{Ba}$ |
| *Dry season* | | | | | | | | |
| Metropolitan (1700) | 25.3 (0.3)$^a$ | 0.45 (0.06)$^{Bb}$ | 5.32 (1.26)$^{bc}$ | 3.42 (1.55)$^{Aa}$ | 82.7 (19)$^B$ | -6.88 (4.14)$^{Ba}$ | 4.18 (4.62) | -4.05 (7.21)$^{Aab}$ |

| | | | | | | | | |
|---|---|---|---|---|---|---|---|---|
| P27 (2030) | 24.7 (0.2)$^{bc}$ | 0.53 (0.08)$^{Bab}$ | 4.46 (0.89)$^{Bc}$ | 0.79 (0.18)$^{Ab}$ | 87.7 (14)$^{B}$ | -12.1 (3.1)$^{Bab}$ | 4.87 (4.70) | 1.09 (1.23)$^{Aab}$ |
| P8 (2360) | 24.9 (0.3)$^{Bab}$ | 0.48 (0.06)$^{Bb}$ | 6.04 (1.15)$^{abc}$ | 3.68 (1.16)$^{Aa}$ | 85.7 (17)$^{B}$ | -21.3 (8.37)$^{Bbc}$ | 5.64 (5.75)$^{B}$ | 6.50 (3.76)$^{Aa}$ |
| P19 (2690) | 25.0 (0.3)$^{ab}$ | 0.49 (0.04)$^{Bb}$ | 7.47 (1.22)$^{ab}$ | 0.64 (0.26)$^{b}$ | 85.5 (12)$^{B}$ | -29.2 (4.08)$^{Bc}$ | 1.30 (3.09)$^{B}$ | -2.41 (2.35)$^{b}$ |
| P32 (3400) | 24.4 (0.3)$^{c}$ | 0.64 (0.09)$^{Ba}$ | 7.86 (1.37)$^{a}$ | 1.17 (0.61)$^{Ab}$ | 78.5 (15)$^{B}$ | -17.4 (5.09)$^{Bab}$ | 5.89 (5.51) | 4.34 (2.23)$^{Aa}$ |

[a] Means ((±SE, $n$ = 4) followed by different lowercase letters indicate significant differences among sites within each season and
different uppercase letters indicate significant differences between seasons within each site (linear mixed effects model with Tukey
HSD test at $P \le 0.05$).
**Table 4** Annual[a] trace gas fluxes (mean (SE), n = 4) from lowland tropical forest soils along
orthogonal gradients of annual precipitation and soil fertility in the Panama Canal watershed,
central Panama.

| Site (annual precipitation) | $CO_2$ | $CH_4$ | $N_2O$ | NO |
|---|---|---|---|---|
| | (Mg C ha$^{-1}$ yr$^{-1}$) | (kg C ha$^{-1}$ yr$^{-1}$) | (kg N ha$^{-1}$ yr$^{-1}$) | (kg N ha$^{-1}$ yr$^{-1}$) |
| Met (1700 mm) | 8.48 (0.70) | -0.34 (0.17) | 0.41 (0.06) | -0.82 (0.16) |
| P27 (2030 mm) | 9.16 (0.62) | -0.51 (0.04) | 0.43 (0.06) | -0.12 (0.04) |
| P8 (2360 mm) | 10.14 (0.76) | -1.45 (0.15) | 1.07 (0.15) | -0.17 (0.17) |
| P19 (2690 mm) | 9.89 (0.49) | -1.98 (0.07) | 0.35 (0.05) | -0.21 (0.10) |
| P32 (3400 mm) | 7.89 (0.84) | -0.94 (0.19) | 0.66 (0.18) | -0.03 (0.09) |

[a] Calculated using the trapezoidal rule between fluxes and time interval, covering the
measurement periods of January - December 2011 for $CO_2$ , $CH_4$ and $N_2O$, and June 2010 - May
2011 for NO. Annual fluxes were not tested statically for differences among sites since these are
trapezoidal extrapolations.

Table 5 Spearman correlations of soil biochemical characteristics[a] and annual (measured in 2011) soil trace gas fluxes from five lowland tropical forests along orthogonal precipitation and fertility gradients in the Panama Canal watershed, central Panama.

| | ECEC | BS | Na | Al | pH | Clay | $CO_2$ | $CH_4$ | $N_2O$ | NO |
|---|---|---|---|---|---|---|---|---|---|---|
| $^{15}$N sig. | -0.87** | -0.67** | -0.30 | 0.42 | -0.61** | -0.15 | 0.41 | -0.70** | 0.30 | 0.16 |
| ECEC | | 0.80** | 0.34 | -0.50 | 0.76** | -0.12 | -0.33 | 0.77** | -0.09 | -0.17 |
| BS | | | -0.13 | -0.87** | 0.96** | -0.12 | -0.40 | 0.78** | -0.12 | -0.54 |
| Na | | | | 0.45 | -0.18 | -0.15 | 0.04 | 0.01 | -0.01 | 0.60** |
| Al | | | | | -0.87** | 0.04 | 0.24 | -0.71** | 0.17 | 0.58** |
| pH | | | | | | -0.04 | -0.34 | 0.76** | -0.12 | -0.54 |
| Clay | | | | | | | -0.13 | -0.17 | -0.67** | -0.34 |
| $CO_2$ | | | | | | | | -0.24 | 0.26 | 0.10 |
| $CH_4$ | | | | | | | | | -0.07 | -0.31 |
| $N_2O$ | | | | | | | | | | 0.19 |

** $P < 0.01$, $n = 20$ (4 replicate plots in each of the 5 forest sites)

[a] Soil parameter abbreviations: $^{15}$N natural abundance signature ($^{15}$N sig.), effective cation exchange capacity (ECEC) and base saturation (BS).

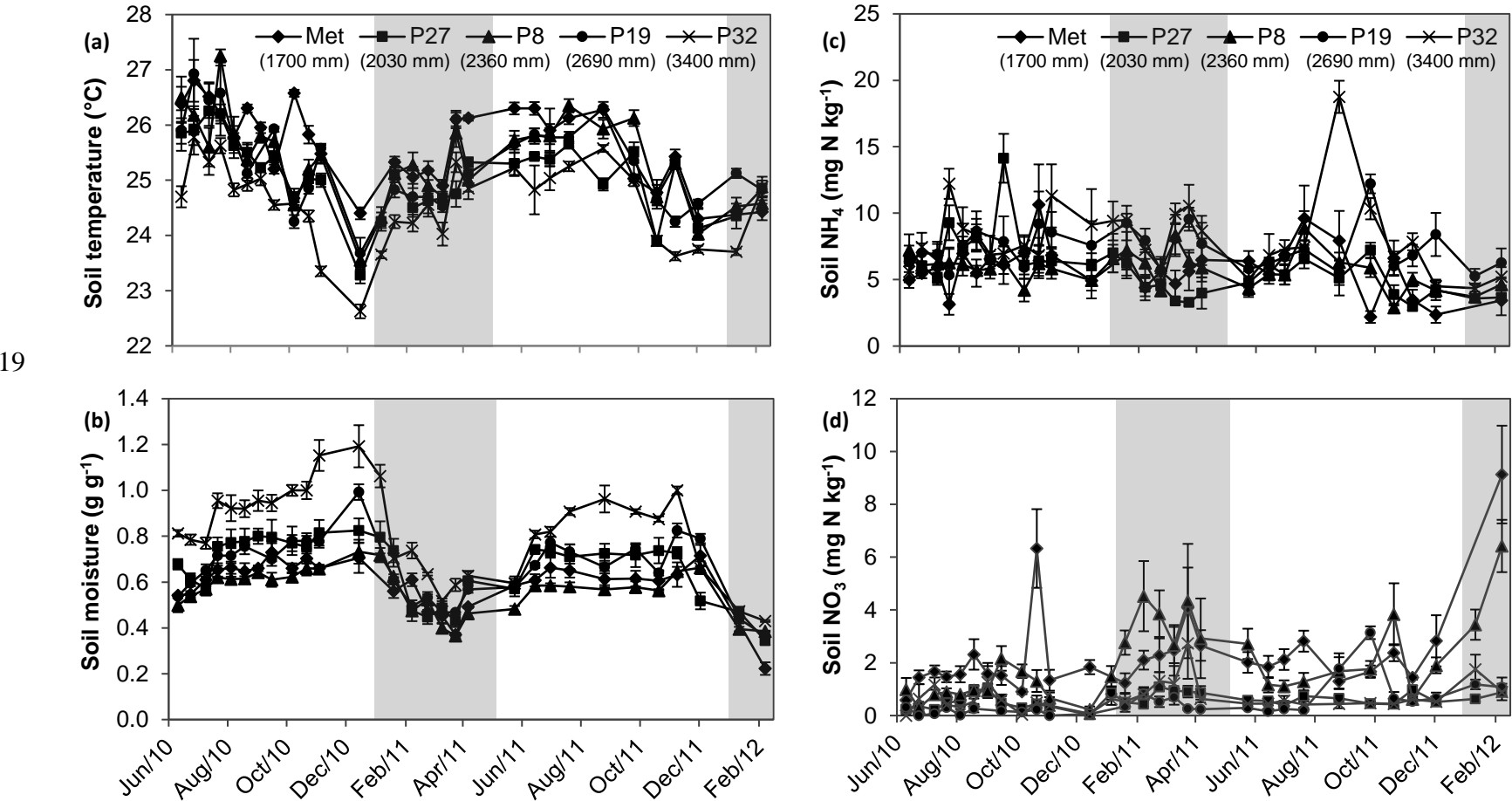

21 **Fig. 1** Mean (±SE, n = 4) soil **(a)** temperature, **(b)** moisture, **(c)** $NH_4^+$ and **(d)** $NO_3^-$ concentrations measured in the top 5 cm of soil in

22 lowland forests along orthogonal gradients of annual precipitation and soil fertility in the Panama Canal watershed, central Panama.

23 Gray shading indicates the dry season (January through April).

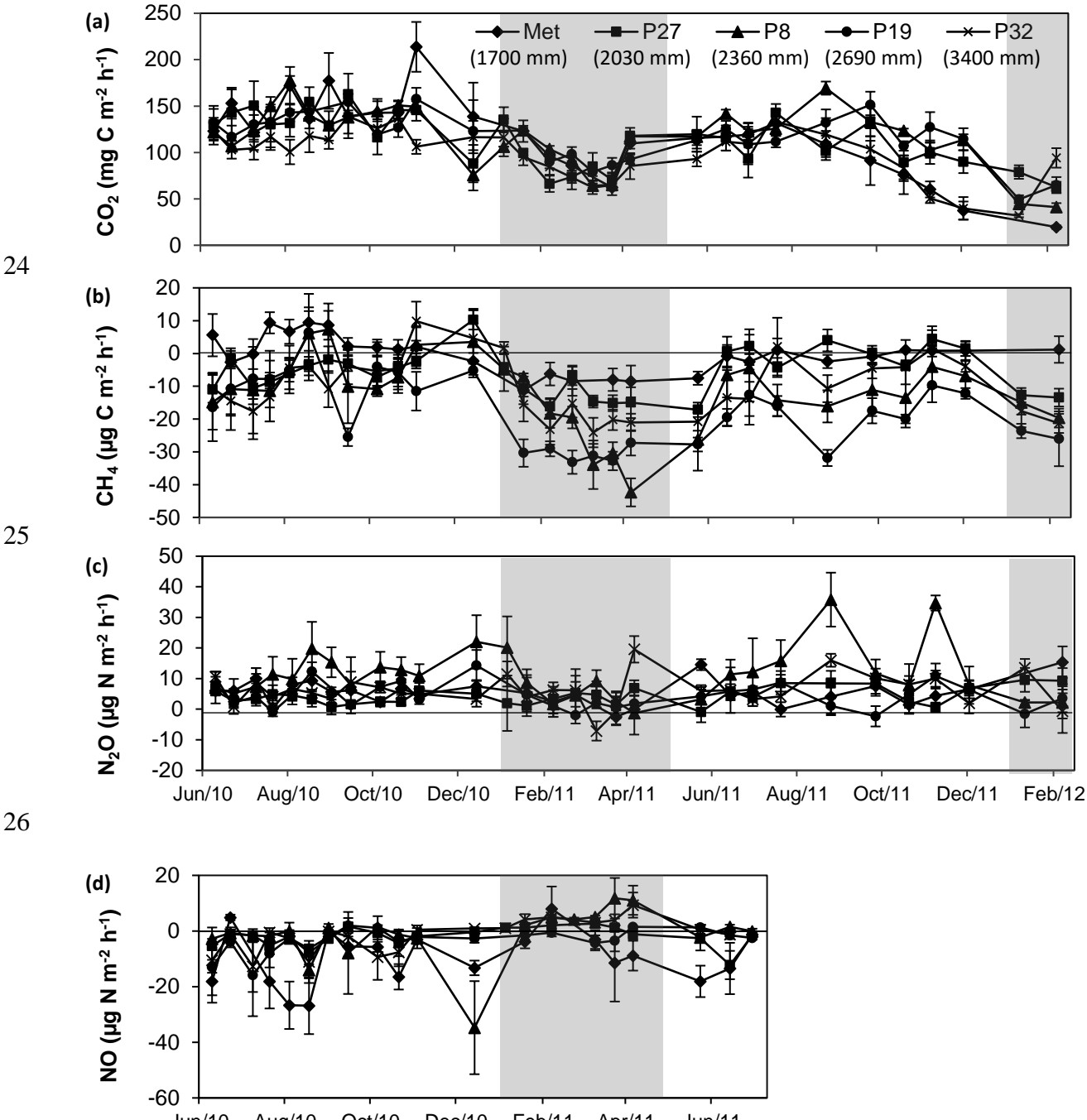

**Fig. 2** Mean (±SE, n = 4) soil **(a)** $CO_2$, **(b)** $CH_4$, **(c)** $N_2O$ and **(d)** NO fluxes from lowland forests

along orthogonal gradients of annual precipitation and soil fertility in the Panama Canal

watershed, central Panama. Gray shading indicates the dry season (January through April).

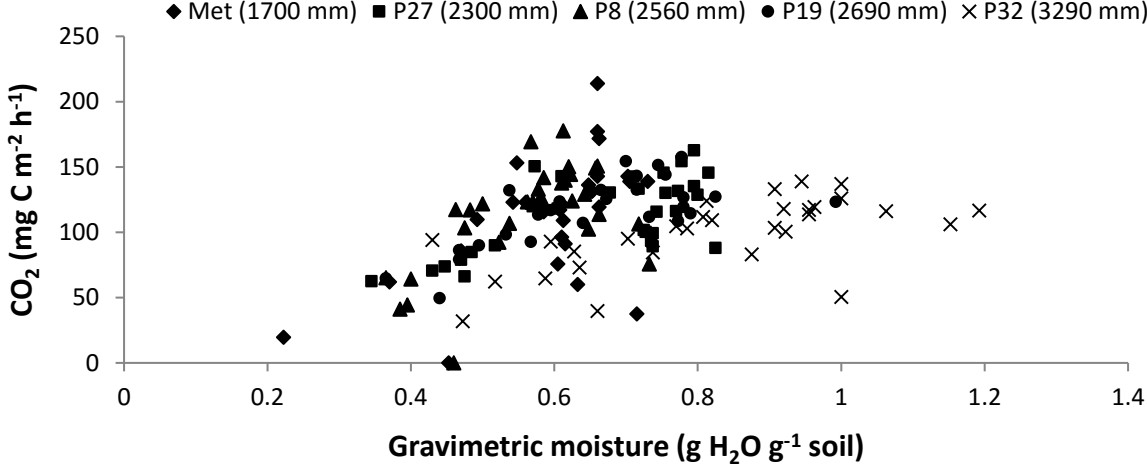

**Fig. 3** Soil $CO_2$ fluxes and moisture contents (top 5 cm) in five lowland forests along orthogonal
gradients of annual precipitation (shown in brackets) and soil fertility in the Panama Canal
watershed, central Panama. Each data point is the average of four replicate plots on one sampling
day from one of the five sites, measured from June 2010 to February 2012 ($n = 145$).

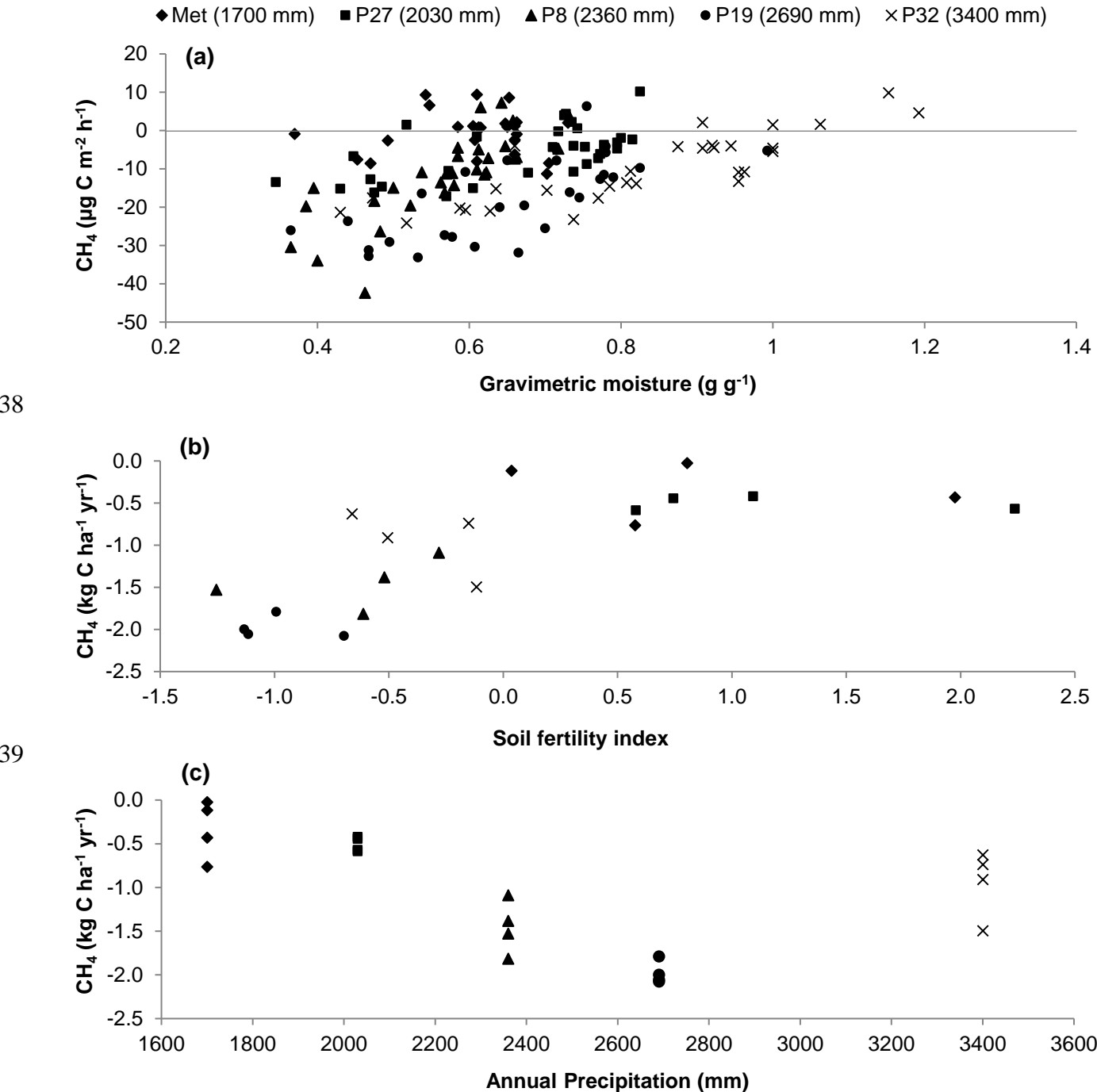

Fig. 4 Average daily soil CH₄ fluxes plotted against (a) soil moisture (top 5 cm), and annual soil CH₄
fluxes plotted against (b) soil fertility index and (c) annual precipitation. For (a), each data point is the
average of four replicate plots on each sampling day of each of the five sites, measured from June 2010
to February 2012. The five lowland forests are located along orthogonal gradients of annual
precipitation and soil fertility in the Panama Canal watershed, central Panama.

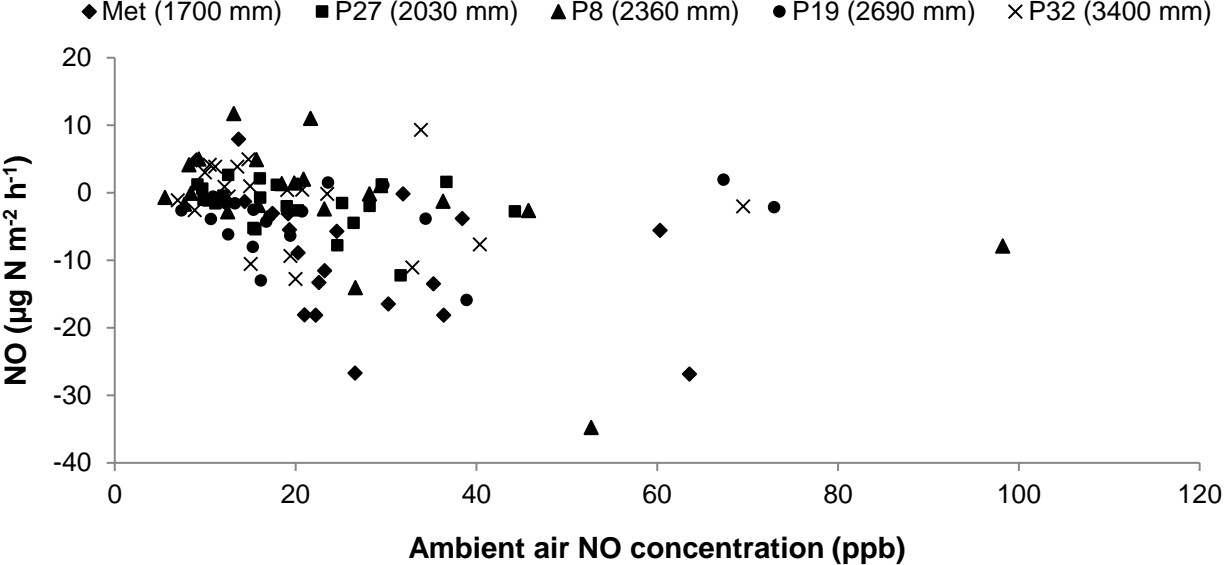

**Fig. 5** Soil NO fluxes plotted against ambient air NO concentrations; each data point is the average of four replicate plots on each sampling day in each of the five sites, measured from June 2010 to June 2011. The five lowland forests are located along orthogonal gradients of annual precipitation and soil fertility in the Panama Canal watershed, central Panama.