# Peer review of "Soil trace gas fluxes along orthogonal precipitation and soil fertility gradients in tropical lowland forests of Panama"

_Biogeosciences, 2016_

## Referee Comment (RC1) · Anonymous Referee #2 · 25 Jan 2017

The manuscript BG-2016-393 titled "Soil trace gas fluxes along orthogonal precipitation and soil fertility gradients in tropical lowland forests of Panama" presents a comprehensive dataset of trace gas fluxes from the tropics and is highly recommended to get published. However, during the review of the manuscript several sections need major improvement before publishing.

General comments: In your data several new aspects are visible, but not presented and discussed in an appropriate way: (1) Since in tropical ecosystems soil moisture is highly variable, while temperature is fairly constant (can be seen in your dataset: while gravimetric soil moisture changed from 1.2 to 0.4,soil temperature changed from 27 to 23°C. In other words 66% change of moisture, while temperature changed 15%.).
Based on that it can be expected that changing soil moisture is the major driver of trace gas emissions. However, in your study a co-correlation of soil moisture and soil temperature is discussed. This is highly interesting, but not yet well presented. You should be able to demonstrate that air temperature at your sites was fairly constant, therefore the most of the change in soil temperature should be attributed to co-correlation to soil moisture changes. Based on theoretical considerations (e.g. $Q_{10}$ value) you should be able to give an estimate about how much of the change in $CO_2$, $CH_4$, $N_2O$ emission could be caused by temperature only and by the combined soil moisture/temperature effect.

(2) The general "parabolic relationship" of $CO_2$ and soil moisture might be influenced by combining all data point from all sites. It seems actually that the emission follow more actual soil moisture than rainfall gradient. For a more comprehensive analysis, it might be helpful to include correlation coefficients for rainfall, soil moisture, soil temperature, $NO_3$- and $NH_4$+. Since in the whole paper all figures show data points with individual symbols for each site, it seems reasonable to use different symbols for each site (Fig3). Furthermore, is there a reason why $N_2O$ is not shown in relationship to soil moisture? It might be helpful for a more process based discussion and the role of aerobic $CH_4$ oxidation coupled to denitrification in this soils? Predominantly the soils are a net-sink for $CH_4$, and you measured $N_2O$ and $NO_3$ but did not discuss the coupling of processes yet (see e.g. Zhu et al. 2016 aerobic methane oxidation coupled to denitrification). It would be more appropriate to convert gravimetric soil moisture into either whc or WFPS to normalize somehow for the soils from different site.

(3) If soil temperature, soil moisture, and soil properties would dominate the $CO_2$, $CH_4$, $N_2O$, and NO fluxes, the data points (Fig.3) should result separate functions over time. The fact, that they are overlaying each other suggests, that other parameters, which are not yet discussed might affect $CO_2$, $CH_4$, $N_2O$, and NO fluxes. As such it should be discussed how abundance (and activity?) of functional microbial groups will change within the rainfall and fertility transect?

[Figure]

(4) Without any additional literature reference the transfer from Tamai et al., 2003 for methanotrophs to methanogens is hard to buy. In Tamai et al., 2003 a negative correlation between CH4 uptake rate and Al was found. Table 2 shows that your inhibition might be possible for P8, P19, P32, but not for the others. However, these 3 sites show actually the lowest CH4 fluxes in the rain season 2011 (Fig. 2). Shouldn't a correlation of net flux and Al result in a positive correlation if inhibition of methanotrophs based on Tamai et al., 2003 is assumed? If your assumption would be valid, how can you explain a simultaneous inhibition of methanotrophs which could cancel out your inhibition of methanogens? Since methanotrophs and methanogens are different functional groups of microbes, I think this is speculative.

(5) For me it seems more plausible that a combination of pH, BS and ECEC which show strong correlations as well, might result a stronger impact for CH4 flux. And a correlation of 15N might point towards coupled methane oxidation and denitrification (e.g. Zhu et al., 2016)? Based on the microbial processes it can be assumed that CH4 oxidation should contribute to CO2 formation. However, this is indicated by a correlation of only -0.24 (CH4 and CO2) in Table 5. Consequently, a potential coupling of aerobic methane oxidation and denitrification might result only -0.07 (CH4 and N2O) in table 5. Finally the introduction and discussion would highly benefit to be focused more on microbial processes. Minor comments: Introduction It might be better for the reader to follow the different microbial processes which cause the production and consumption of each trace gas rather than jump from effects of temperature to moisture to soil properties on CO2, CH4, N2O and NO? Overall the introduction is missing a clear structure. You are writing about methanotrophs and methanogens, but for the other trace gases you don't include any information about the processes and functional microbial groups. Line 40: Studies (without references) either include references or refer to a comprehensive list in supplement. Line 65/66: take care of terminology, maybe define once? Net CH4 flux consists of production (positive) and consumption (negative). Furthermore, it should be mentioned that production occurs even under negative net CH4 flux, but consumption is predominant.

Material and Methods Line 149 "soil trace gas flux measurement": you can only measure mixing ratios. Fluxes are the result of a second order calculation. Line 150 "fluxes were measured"? Line 168 Please specify what gas did you flow through the chambers? Ambient air, synthetic air? I recommend including the formulas to calculate $CO_2$, $CH_4$, $N_2O$ (static) and NO (dynamic), plus the trapezoid rule to calculate the annual fluxes that the reader does not have to look up several other papers to follow the calculations. Results The results are majorly focusing on the descriptive correlations. Why the major results of $CO_2$, $CH_4$, $N_2O$, NO fluxes is not presented here? For me these are the major results obtained from the field by hard work (Fig1 and Fig2). Line 291 Due to different soil properties for each site, it seems not very helpful to present Fig. 3 and talk about a "parabolic relationship".

Discussion Statement about what might cause the $NO_3-$ differences? Wet deposition, if yes, are there values from literature? The connection of the trace gas fluxes to microbial processes is missing. E.g. the correlation of $CH_4$ fluxes (net uptake) is negatively correlated to 15N natural abundance. Does this point towards a $CH_4$ production coupled to denitrification? And could this coupling be less relevant in the dry season versus the wet season and thereby result amplified correlations in the dry season?

Figures: Error bars are missing for Fig 3, 4, and 5 Fig. 4 a, b, c should include a 0 line for easier understanding. Fig. 4a might be better to bin data into moisture classes of 10%. Less data points will make the figure easier to understand and better show trends. Error bars can be included. Would it make more sense to average the single points and report error bars to highlight the grouping in different fertilizer regimes Fig 4b? That might be helpful for discussion? Fig. 5: Where was the NO ambient mixing ratio measured? Close to the ground (chamber height) or 2m height? Are there references available for such high NO ambient mixing ratios and possible sources? Based on Remde et al (1989) it might be helpful to plot NO release rate versus ambient NO mixing ratio at same moisture and temperature for each site. Furthermore, only data points for a range of soil moisture and soil temperature should be selected.

Please also note the supplement to this comment:
http://www.biogeosciences-discuss.net/bg-2016-393/bg-2016-393-RC1-supplement.pdf

---

## Referee Comment (RC2) · Y. A. Teh (Referee) · 27 Jan 2017

**GENERAL COMMENTS**

This is an elegantly written paper, that nicely frames the research within the wider theoretical context of soil trace gas fluxes in the tropics, and succinctly reviews our current understanding and identifies major knowledge gaps. Importantly, the author correctly identifies the fact that while existing theory about the controls on soil trace gas exchange in the tropics are underpinned by theoretical expectations, there have been relatively few tests of this theory across natural rainfall or fertility gradients in the tropics. The work presented here tests and further develops our theoretical understanding by employing these natural gradients to tease-out the synergistic effects of

rainfall, nutrients, and other environmental drivers on modulating soil-atmosphere trace gas exchange. The data presented here are thus ideal not only for assessing theory, but could also for testing, developing and parameterising bottom-up process models, such as DNDC or DAYCENT.

In addition to the general contributions that this paper makes, I believe it also helps to enhance our understanding of specific biogeochemical cycles. For example, this paper helps to develop our understanding of the multiple constraints placed on methanotrophy in tropical soil. Prior studies have tended to focus on a smaller sets of control variables (e.g. moisture and temperature only), whereas this study takes a more comprehensive look at the role of environmental variables and seasonality on methane uptake.

Likewise, this paper enhances our understanding of less well-studied gases such as nitric oxide (NO), for which we know far less than it's more "popular" sister compound nitrous oxide (N2O). The overall trend towards net NO uptake across this area is intriguing, given that past studies in CSA has emphasised the role of these systems as regional/global NO emission sources. It would be useful if the authors could develop or speculate on the wider implications of NO uptake for local and regional atmospheric chemistry, as I do not believe that our wider community has fully engaged with the notion of soils as NO sinks, given the past emphasis on soils as NO emission sources.

While I did not have major criticisms of this paper, I do have a few suggestions for improvement. First, it would be useful if the authors could make more use of multiple regression or the mixed effects models to determine the hierarchy (i.e. relative importance) of environmental drivers for different trace gases (i.e. which are the dominant and which are the lesser environmental controls?). While the authors have outlined the dominant role of soil moisture, it would be interesting to see a clearer description of the relative importance of the other drivers. Does the hierarchy of drivers vary among sites? Do the hierarchy of drivers vary among seasons? This would help develop not only a more "global" understanding of which drivers are dominant in this region, but

also help us understand how individual study sites differ from each other at different times of year.

Second, in the section on soil CO2 flux, I think it would be useful if the authors could revise the text to incorporate a slightly expanded discussion of how root respiration could be influencing variations in soil CO2 fluxes (see point 10 below). For example, could the differences in respiration between this study site and others be attributed to differences in belowground biomass or root/shoot allocation? Do data exist on belowground biomass in these sites? If so, do those data help explain patterns in soil respiration?

Third, in section 4.2 of the Discussion, the authors have identified separate sets of controls on CH4 uptake that appear to be operating on different time scales; i.e., daily fluxes of CH4 appear to be more strongly linked to soil moisture, whereas soil fertility was a stronger constraint on annual CH4 fluxes. This is an important and interesting finding, as it highlights the scale-dependency of different environmental controls, and suggests that different environmental factors may be controlling different aspects/components of trace gas cycling; e.g. in the short-term, soil moisture may be regulating transport and supply of CH4 to methanotrophs (hence, regulating instantaneous fluxes), whereas in the long-term, site fertility may be influencing the total amount of methanotrophic biomass or the overall methanotrophic potential of these soils. It would be useful if the authors could consider a way of revising the current text to better highlight this important finding, as it has wider implications for upscaling these results or incorporating these findings into process-based models.

Other minor specific comments are provided below.

**SPECIFIC COMMENTS**

1. Lines 158-165: It would be useful know the precision of the analysis; i.e. what was the coefficient of variation for the standards?

2. Lines 173-174: Ibid.

3. Lines 175-177: Were any fluxes non-linear? How were these data treated? Under more saturated soil moisture conditions, was there any evidence of ebullition? If so, how were these data treated?

4. Lines 205-207: Are there any potential limitations associated with using this 15N natural abundance technique?

5. Lines 254-256: Have the authors considered using Box-Cox transformations to normalise the data? If successful, this would enable the authors to use parametric statistics (e.g. linear regression, multiple regression) rather than Spearman's Rank correlation. Moreover, even if the data do not fully meet the assumptions for parametric analyses, it may be useful/instructive to analyse the data using multiple regression techniques to evaluate the relative hierarchy of environmental drivers.

6. Lines 271-286: It is worthwhile reporting the seasonal trends (or, lack of trends in NH4+) here as well. Does NH4+ show wet or dry season differences? I had assumed not given that this wasn't stated explicitly. This is interesting because if there are no significant differences, which could be interpreted that rates of net ammonification/NH4+ mineralisation are relatively invariant (although of course you cannot infer whether this is due to invariance in N mineralisation or DNRA rates).

7. Lines 274-275: Do you have complementary measurements of net or gross N cycling processes to help interpret these field patterns? It's possible that the reduction in NO3during the wet season may be linked to reduced nitrification (with a growth of anoxic microsites), or an increase in NO3- reduction (e.g. DNRA or denitrification).

8. Lines 293-297: Were these data from bivariate regressions or from a multiple regression model? If the second, it would be useful to indicate, based on the sum of squares, which variables accounted for a larger proportion of the variance and which variables accounted for less, in order to clearly establish the hierarchy of drivers.

9. Lines 317-319: Increased evidence for nutrient limitation of methanotrophy? What

are the implications of this for process models (could be discussed in the Discussion)?

10. Lines 339-341: Evidence for very active nitrifiers? Perhaps this could be explored further in the discussion.

11. Lines 362-363: To what extent is inter-annual variability modified/affected by differences in belowground allocation and variations in root-rhizosphere respiration? Do data exist on the belowground biomass across your gradient or differences in root/shoot allocation? If so, this may help tease-out the extent to which differences in total soil respiration are affected by differences in the fluxes from individual respiration components.

---

## Author Response (AR1)

Specific response to reviewer comments (including line numbers for the no-markup version)
Reviewer 1
*It would be useful if the authors could develop or speculate on the wider implications of NO*
*uptake for local and regional atmospheric chemistry, as I do not believe that our wider*
*community has fully engaged with the notion of soils as NO sinks, given the past emphasis on*
*soils as NO emission sources.*
We agree that our observation of regular NO uptake by soils is worth exploring, given that until
now the majority of studies have reported soils acting as net NO sources. However, we do not
want to overemphasize this point, as our soils experienced unique ambient conditions (i.e. high
atmospheric NO and O3) that may not occur in many other sites worldwide. However, in
response to this comment, we have revised this paragraph in the discussion to emphasize the
importance of these unusual results. The final sentence of Section 4.4 (Line 599-603) now reads:
In summary, although the soils in our study sites can be a net source of NO, particularly during
the dry season (Fig. 2d) and in sites where ambient air NO concentrations are low (Fig. 5), most
of the time the soils acted as net sinks of NO, signifying the importance of soil and vegetation as
NO sinks (Jacob and Bakwin, 1991; Sparks et al., 2001) in areas affected by anthropogenic NO
sources.
*First, it would be useful if the authors could make more use of multiple regression or the mixed*
*effects models to determine the hierarchy (i.e. relative importance) of environmental drivers for*
*different trace gases (i.e. which are the dominant and which are the lesser environmental*
*controls?). While the authors have outlined the dominant role of soil moisture, it would be*
*interesting to see a clearer description of the relative importance of the other drivers. Does the*
*hierarchy of drivers vary among sites? Do the hierarchy of drivers vary among seasons?*
In response to this comment and comments from the other reviewer (see below), we added Table
S1, which shows the hierarchy of importance of the soil factors controlling soil GHG fluxes for
each season (across sites) and within each site (across seasons).
*Table S1 Ranking[a] of soil factors that control the soil-atmosphere trace gas exchange along*
*orthogonal precipitation and fertility gradients in the Panama Canal watershed, central Panama*
*(F- and P-value of the model ANOVA shown in brackets).*

| | $CO_2$ | $CH_4$ | $N_2O$ | NO |
|---|---|---|---|---|
| Wet season (all sites) | 1. $NH_4^+$ (F=24.5, P<0.01)
 2. Temperature (F=9.4, P<0.01) | 1. Moisture (F=59.1, P<0.01)
 2. Temperature (F=10.0, P<0.01) | 1. $NO_3^-$ (F=6.1, P=0.01) | ns |

| | | | | |
|---|---|---|---|---|
| | | 3. $NO_3^-$ (F=5.6, P=0.02) | | |
| Dry season (all sites) | 1. Moisture (F=52.4, P<0.01) 2. Temperature (F=5.01, P=0.03) | 1. Moisture (F=10.5, P<0.01) 2. $NO_3^-$ (F=14.6, P<0.01) 3. $NH_4^+$ (F=7.8, P<0.01) | ns | ns |
| Met (wet/dry) | 1. Moisture (F=38.0, P<0.01) 2. $NH_4^+$ (F=13.3, P<0.01) | ns | ns | ns |
| P27 (wet/dry) | 1. Temperature (F=25.9, P<0.01) 2. Moisture (F=22.7, P<0.01) | 1. Moisture (F=33.1, P<0.01) 2. Temperature (F=5.2, P=0.03) | ns | 1. Temperature (F=10.1, P<0.01) 2. Moisture (F=7.4, P<0.01) |
| P08 (wet/dry) | 1. Moisture (F=25.8, P<0.01) 2. Temperature (F=20.6, P<0.01) | 1. Moisture (F=30.8, P<0.01) | 1. Moisture (F=12.8, P<0.01) | 1. Moisture (F=16.6, P<0.01) |
| P19 (wet/dry) | 1. Moisture (F=44.2, P<0.01) 2. $NH_4^+$ (F=4.2, P=0.04) | 1. Moisture (F=32.5, P<0.01) | 1. Moisture (F=27.7, P<0.01) 2. $NO_3^-$ (F=14.2, P<0.01) | ns |
| P32 (wet/dry) | 1. Moisture (F=18.8, P<0.01) 2. Temperature (F=16.0, P<0.01) 3. $NO_3^-$ (F=4.2, P=0.04) | 1. Moisture (F=62.5, P<0.01) 2. $NH_4^+$ (F=7.8, P<0.01) | 1. Moisture (F=7.2, P<0.01) | ns |

[a] This ranking (denoted by numbers) signifies its hierarchy of importance based on the minimal adequate
LME model, using a stepwise model simplification; ns – no soil factor showed significant relationship
with the soil trace gas fluxes.

*Second, in the section on soil $CO_2$ flux, I think it would be useful if the authors could revise the*
*text to incorporate a slightly expanded discussion of how root respiration could be influencing*
*variations in soil CO2 fluxes (see point 10 below). For example, could the differences in*
*respiration between this study site and others be attributed to differences in belowground*
*biomass or root/shoot allocation? Do data exist on belowground biomass in these sites? If so, do*
*those data help explain patterns in soil respiration?*

We do not know of any data existing on root biomass in any of our present sites. From our
previous work and that of others, we know root respiration can contribute 30% - 35% of the soil
$CO_2$ efflux (van Straaten et al. 2011, Silver et al. 2005). However, we do not have any root data
to base any possible contribution of roots to the soil $CO_2$ fluxes at our present sites. Interestingly,
regardless of the contribution of autotrophic respiration to the soil $CO_2$ fluxes, we did not detect
any significant differences in soil $CO_2$ fluxes among sites, but only found that across our 5 sites
the temporal pattern of soil $CO_2$ fluxes was strongly related to soil moisture contents (Fig. 3)
[added at line 424-427]. The range of soil moisture contents in these 5 sites (Fig. 4a) also clearly
showed that the low-rainfall sites varied from the lower end up to the mid-moisture range, the
high-rainfall site varied from the mid to high-moisture ranges and there was a wide overlap
among sites within the mid-moisture ranges (Fig. 4a). Thus, if both autotrophic and heterotrophic
responded similarly to these ranges of soil moisture contents, then their relative contributions
should be less important than their overall response, or the response of soil $CO_2$ fluxes as a
whole, to soil moisture contents.

Thus, in order to avoid any unnecessary speculative discussion, we prefer to focus our discussion
on the possible causes of the generally low soil $CO_2$ fluxes from our present sites as compared to
the other lowland forests in Panama (lines 406-423) – possibly due to low root respiration as well
as considerable variation in litterfall (as a substrate for heterotrophic respiration). Here, we
speculate that autotrophic respiration could be low at two of our sites since they have lower tree
densities (particularly at Met and P27; see 2.1) than the old growth, lowland forests on BCI and
Gigante.

To summarize, we focused our discussion on the temporal pattern, which our data have clearly
shown, as well as reporting on similarities and differences with other studies from CSA lowland
forests.

*Third, in section 4.2 of the Discussion, the authors have identified separate sets of controls on*
*CH4 uptake that appear to be operating on different time scales; i.e., daily fluxes of CH4 appear*
*to be more strongly linked to soil moisture, whereas soil fertility was a stronger constraint on*
*annual CH4 fluxes. This is an important and interesting finding, as it highlights the scale-*
*dependency of different environmental controls, and suggests that different environmental*
*factors may be controlling different aspects/components of trace gas cycling; e.g. in the short-*
*term, soil moisture may be regulating transport and supply of CH4 to methanotrophs (hence,*
*regulating instantaneous fluxes), whereas in the long-term, site fertility may be influencing the*

*total amount of methanotrophic biomass or the overall methanotrophic potential of these soils. It*
*would be useful if the authors could consider a way of revising the current text to better highlight*
*this important finding, as it has wider implications for upscaling these results or incorporating*
*these findings into process-based models.*
We have made major revisions to Section 4.2 in order to highlight better the scale-dependency of
different environmental drivers of soil $CH_4$ fluxes. See Lines 464-510.
Additional comments
*1. Lines 158-165: It would be useful know the precision of the analysis; i.e. what was the*
*coefficient of variation for the standards?*
*2. Lines 173-174: Ibid.*
We added the detection limits of our instruments, which were calculated as 3 x standard error of
the standard, which was used to check the instrument precision during the analysis. The average
detection limit during the periods of our measurements was 50 ppm $CO_2$, 43 ppb $N_2O$, 45 ppb
$CH_4$, and 0.04 ppb NO/mV (mV is the electrical signal from the produced chemiluminescence of
the oxidized NO). (Lines 174-175, 194-195)
*3. Lines 175-177: Were any fluxes non-linear? How were these data treated? Under more*
*saturated soil moisture conditions, was there any evidence of ebullition? If so, how were these*
*data treated?*
Soil NO fluxes were always linear. We show below the typical soil NO fluxes where we
observed net emission and net consumption. We considered the first 3-min. of linear change in
NO concentrations with chamber closure time. (added at Lines 198-204)
For soil $CO_2$, $N_2O$ and $CH_4$fluxes, all 3 gases were analyzed in our gas chromatograph
sequentially from the same gas sample. Since these 3 gases come from the same sample, we
based our best fit of gas concentration vs. time on the $CO_2$ concentration increase, as it is the gas
with the highest concentration among these 3 gases. The $CO_2$ concentration always increased
linearly with time of chamber closure. Hence, we used a linear fit for all the 3 gases, and zero
fluxes and negative fluxes (i.e. for $N_2O$ and $CH_4$) were all included in our data analysis. [added
at Lines 198-204] This linear increase was not surprising, considering that the large volume of
our chambers (11 L) decreases the likelihood of feedbacks on the diffusion gradient with
increasing concentration; additionally, there was generally low soil $CO_2$ and $N_2O$ fluxes at our
sites (as we noted in the Discussion).
We also did not observe any evidence of ebullition (e.g. sudden increase of gas concentration
during our 30-min chamber closure). (mentioned in Lines 198-204) Such a phenomenon is more
likely under flooded conditions or in the transition periods to and from flooded conditions. We
measured the volumetric moisture content continuously in our wettest site (plot 32) during the
study period, using permanently installed water content probes (Campbell Scientific CS616,
Logan, Utah), the same instrument we describe in our earlier studies in another lowland forest in
Gigante, Panama (Koehler et al., 2010, Veldkamp et al. 2013, and Corre et al. 2014). The water- filled pore space in the top 10 cm, recorded in the data logger every 4 hours, did not reach
saturation at any time during our measurement period.

[Figure]

(a) Plot 8 on June 22, 2010 (showing net NO emission, Fig. 2d of the manuscript)

[Figure]

(b) Metropolitano on June 16, 2010 (showing net NO consumption, Fig. 2d of the manuscript)

Time (minute) of measurements, recorded in a data

4. Lines 205-207: Are there any potential limitations associated with using this 15N natural
abundance technique?

The potential limitations of using 15N natural abundance of the soil is its inherent high spatial variability brought about by 1) vegetation species differences, and 2) surface topography, which may drive differences in soil 15N natural abundance due to slope influences on water and solute distribution and ultimately on microbial N-cycling processes. These are the reasons why we used not only the 15N natural abundance of the surface depth but also of the 4 depth increments, and determined the overall 15N nat. abund. enrichment factor ($\varepsilon$), which considers the change in 15N natural abundance signature and total N concentration with depth in relation to the surface depth, as an integrative indicator of soil N availability (shown in our previous studies, e.g. Baldos et al. 2015). [summarized in Lines 233-236]

*5. Lines 254-256: Have the authors considered using Box-Cox transformations to normalise the data? If successful, this would enable the authors to use parametric statistics (e.g. linear regression, multiple regression) rather than Spearman's Rank correlation. Moreover, even if the data do not fully meet the assumptions for parametric analyses, it may be useful/instructive to analyse the data using multiple regression techniques to evaluate the relative hierarchy of environmental drivers.*

We have added an additional table (Table S1 above) to show the relative hierarchy of environmental drivers within sites (across seasons) and within seasons (across sites). Non-parametric statistics were only used to compare non-repeated measures with annual and seasonal averages.

*6. Lines 271-286: It is worthwhile reporting the seasonal trends (or, lack of trends in NH4+) here as well. Does NH4+ show wet or dry season differences? I had assumed not given that this wasn't stated explicitly.*

Lines 309-310 state that of the four repeated measures (temperature, moisture, extractable $NO_3^-$ and extractable $NH_4^+$), only moisture and extractable $NO_3^-$ exhibited strong seasonal differences. Additionally, we have added a statement (line 313-315) specifically clarifying that temperature and extractable $NH_4^+$ exhibited between-season differences at only one site each (temperature - P8, extractable $NH_4^+$ - P27).

*7. Lines274-275: Do you have complementary measurements of net or gross N cycling processes to help interpret these field patterns? It's possible that the reduction in NO3 during the wet season may be linked to reduced nitrification (with a growth of anoxic microsites), or an increase in NO3- reduction (e.g. DNRA or denitrification).*

These lines that the reviewer is referring to are presenting the total soil N, which is commonly 4 orders of magnitude (Table 2) higher than the mineral N (Table 3), the latter reflecting the actively cycling fraction of the total N. Thus, the rate of soil N cycling, being small compared to the total soil N, cannot make a big change to the amount of total N. Total soil N reflects the long-term accumulation of N in these sites.

We indeed have measured gross rates of soil-N cycling in the same sites and replicate plots in the wet season 2010 (Nov.) and the dry season 2011 (May). We do not report them here, as they are included in a separate paper focusing on patterns of soil-N cycling and soil N availability along
these orthogonal gradients of soil fertility and precipitation.
However, our interpretations in the present paper were considered in light of the rates of soil-N
cycling that we measured. Across sites, gross N mineralization rates correlated with soil
microbial biomass N, total soil N, 15N nat. abund. enrichment factor ($\epsilon$), and 15N nat. abundance
(Spearman rank correlation coefficients of 0.48-0.80, n=20, P<0.05). The patterns of microbial N
and total N followed that of increasing annual precipitation, while 15N nat. abund. enrichment
factor and 15N nat. abundance were low at the low- and high-rainfall sites and peaked at the
mid-rainfall sites (Table 2). These patterns were opposite to those of soil pH, ECEC and
exchangeable bases across sites (higher values at the low-rainfall sites with less-weathered soils
than at the mid- and high-rainfall sites with highly weathered soils; all $P \leq 0.05$; Tables 2). Thus,
our interpretation of this pattern of total soil N with increasing precipitation was that the higher
the total N (with increasing precip.), the higher the amount of microbial N and the higher the soil
N availability, as indicated by the rate of actively cycling N (i.e. gross rates of N mineralization)
and mineral N (i.e. soil $NH_4^+$ levels, Table 3).
Across sites and seasons, gross N mineralization was not correlated with gross nitrification but
instead with $NH_4^+$ immobilization (suggesting that heterotrophic nitrification was possibly
important rather than autotrophic nitrification). We cannot merely attribute the reduction of
$NO_3^-$ in the wet season to reduced nitrification because gross nitrification was only measured
once in the wet and once in the dry season, and we did not see significant differences between
wet and dry seasons across sites nor at each site. Additionally, gross nitrification was correlated
with $NO_3^-$ immobilization, but not with DNRA, suggesting that when there was high $NO_3^-$
availability, this was preferably assimilated by the microbial biomass. On the other hand, the soil
$NO_3^-$ levels we show in Table 3 were measured repeatedly, parallel to soil trace gas flux
measurement, over our 21-month study period (as opposed to the gross rate of soil-N cycling
which, due to the intensive labor and cost required, was only measured twice). The soil $NO_3^-$
levels (Table 3) reflected the concurrently occurring $NO_3^-$ production and consumption
processes, [included in Lines 534-543] and our discussion on the role of soil $NO_3^-$ levels on the
soil trace gas fluxes always considered the soil $NO_3^-$ patterns between seasons, among sites, the
inverse correlations of $NO_3^-$ and soil moisture, and the correlations of $NO_3^-$ with soil $CO_2$,
$CH_4$, $N_2O$ and NO fluxes at a particular site.
*8. Lines 293-297: Were these data from bivariate regressions or from a multiple regression*
*model? If the second, it would be useful to indicate, based on the sum of squares, which*
*variables accounted for a larger proportion of the variance and which variables accounted for*
*less, in order to clearly establish the hierarchy of drivers.*
As shown above, we have included a table (Table S1) showing the relative hierarchy of
environmental drivers. As suggested by reviewer 2 above, we used the minimum adequate LME
models in analyzing the hierarchy of drivers. This statistical analysis is also described in the
revised manuscript (line 282-296).
*9. Lines 317-319: Increased evidence for nutrient limitation of methanotrophy? What are the*
*implications of this for process models (could be discussed in the Discussion)?*

This question of the reviewer is related to the 3$^{rd}$ general comment above (please see our answer
above as well). The sentences following these lines 317-319 presented the possible reasons
(through correlations with controlling factors) for this pattern of differences among sites (see
lines 318-331) and are discussed in lines 438-459 of the original manuscript.
The most important controlling factor on the long-term pattern of soil $CH_4$ fluxes across sites
was soil fertility. Specifically, as shown by the strong inverse correlation between soil 15N
natural abundance signatures and exchangeable cations (Table 5), the positive correlation
between soil CH4 flux and fertility (Fig. 4b) likely reflected the long-term effects of soil
development (Tables 1 and 2) - more CH4 uptake occurred in highly weathered soils with less
rock-derived nutrients but high soil N availability (i.e. high 15N natural abundance signatures)
(Tables 4 and 5). When separated by season, the correlation between average soil CH4 fluxes
and soil 15N natural abundance was stronger in the dry season than the wet season (Table S2),
supporting our claim that soil N availability enhanced CH4 uptake in soils when gas diffusion
was favorable (dry season). (included in revised 4.2; See Lines 464-510)
*10. Lines 339-341: Evidence for very active nitrifiers? Perhaps this could be explored further in*
*the discussion.*
This question is related to comment #7 above (please refer to our extended answer there). Our
measured gross nitrification rates (measured once in the wet and once in the dry season at all
sites) did not show significant differences among sites nor between seasons at each site. Thus,
we cannot simply attribute the results presented in these lines (339-341) that the reviewer is
asking (i.e. positive correlations between  soil N2O emissions and moisture and negative
correlations between soil N2O emissions and NO3- concentrations at the mid-rainfall sites (P8
andP19) to be due to active nitrifiers.
*11. Lines 362-363: To what extent is inter-annual variability modified/affected by differences in*
*belowground allocation and variations in root-rhizosphere respiration? Do data exist on the*
*belowground biomass across your gradient or differences in root/shoot allocation? If so, this*
*may help tease out the extent to which differences in total soil respiration are affected by*
*differences in the fluxes from individual respiration components.*
This question is related to the question about $CO_2$ above (please refer to our extended answer
there). In brief, we do not know of any available datasets that could answer this question.
However, we do think our results highlight another interesting facet of $CO_2$ emissions in these
sites, namely, that despite the differences in soil factors between sites, we did not see differences
in $CO_2$ fluxes. However, we did see strong temporal patterns, and therefore  focused our
discussion on short-term changes over time, as well as reporting on similarities and differences
with other studies from CSA lowland forests.
Reviewer 2
*(1) Since in tropical ecosystems soil moisture is highly variable, while temperature is fairly*
*constant (can be seen in your dataset: while gravimetric soil moisture changed from 1.2 to*

*0.4, soil temperature changed from 27 to 23°C. In other words 66% change of moisture, while*
*temperature changed 15%.). Based on that it can be expected that changing soil moisture is the*
*major driver of trace gas emissions. However, in your study a co-correlation of soil moisture*
*and soil temperature is discussed. This is highly interesting, but not yet well presented. You*
*should be able to demonstrate that air temperature at your sites was fairly constant, therefore*
*the most of the change in soil temperature should be attributed to co-correlation to soil moisture*
*changes. Based on theoretical considerations (e.g. Q10 value) you should be able to give an*
*estimate about how much of the change in CO2, CH4, N2O emission could be caused by*
*temperature only and by the combined soil moisture/temperature effect.*

In response to this comment and the first comment of reviewer 2 (see above), we have used
minimal adequate linear mixed effects models to identify a hierarchy of importance of the
environmental drivers within and between sites/seasons, which addresses this concern by giving
more details about the relative importance of moisture, temperature and extractable mineral N
over our 21-month measurement period (Table S1).

*(2) The general "parabolic relationship" of CO2 and soil moisture might be influenced by*
*combining all data point from all sites. It seems actually that the emission follow more actual*
*soil moisture than rainfall gradient. For a more comprehensive analysis, it might be helpful to*
*include correlation coefficients for rainfall, soil moisture, soil temperature, NO3- and NH4+.*
*Since in the whole paper all figures show data points with individual symbols for each site, it*
*seems reasonable to use different symbols for each site (Fig3).*

In response to this comment, we include below a revised version of Fig. 3 to show individual
symbols for each site, which we also use to replace the previous version of Fig. 3 in the
manuscript. As we discussed in lines 440-441, such parabolic relationships have also been
observed by other studies in tropical forests of Costa Rica, Panama and Brazil. Indeed, we
conducted correlation tests between annual soil CO2 emissions and annual rainfall but this was
not statistically significant because their relationship was parabolic and not linearly correlated.
For this parabolic relationship with annual rainfall, we decided it was better to present Fig. 3 as
essentially the same pattern is depicted, but Figure 3 is better, as it depicts the actual measured
daily values. All the other soil factors were tested for correlation with soil CO2 fluxes, and we
have reported their relationships in lines 332-340. They are also summarized in Table S1 (see
above).

[Figure]

*Furthermore, is there a reason why N2O is not shown in relationship to soil moisture? It might
be helpful for a more process based discussion and the role of aerobic CH4 oxidation coupled to
denitrification in this soils? Predominantly the soils are a net-sink for CH4, and you measured
N2O and NO3 but did not discuss the coupling of processes yet (see e.g. Zhu et al. 2016 aerobic
methane oxidation coupled to denitrification).*

As we mentioned in the text (lines 380-383), soil moisture was only strongly correlated with soil
$N_2O$ fluxes at two sites (P8 and P19), so including a figure showing the relationship with
moisture across sites would not add anything to the results we were presenting. We have, indeed,
analyzed our data to explore whether there was a link between soil N2O and CH4 fluxes – which
would support what reviewer 1 is asking here. However, neither Table 5 nor Table S2 support
such link (i.e. no significant correlation). Thus, we would have no basis to include this aspect in
our discussion without being overly speculative. We do note, though, that the roles of NO3- on
soil N2O fluxes as well as on CH4 uptake were discussed extensively (lines 482-492 and 530-
of the manuscript).

*It would be more appropriate to convert gravimetric soil moisture into either whc or WFPS to
normalize somehow for the soils from different site.*

We are unable to convert the gravimetric to WFPS mainly because the Metropolitan and P27
sites, whose parent materials are agglomerates, had fine stones making measurement of soil bulk
density erroneous. We tried to do a more accurate estimate of soil bulk density but we were not
confident that we were able to get out all of the gravel in these heavy clay soils (average soil
texture within the top 50 cm was 60-62%) from the soil cores we used to measure soil bulk
density. On the other hand, we made very careful measurements of gravimetric measure contents
every time we took subsamples from the soils that were concurrently sampled during each soil
gas flux measurements. Thus, our gravimetric moisture measurements were more reliable than
converting to WFPS.

*(3) If soil temperature, soil moisture, and soil properties would dominate the CO2, CH4, N2O,
and NO fluxes, the data points (Fig.3) should result separate functions over time. The fact, that*

*they are overlaying each other suggests, that other parameters, which are not yet discussed*
*might affect CO2, CH4, N2O, and NO fluxes. As such it should be discussed how abundance*
*(and activity?) of functional microbial groups will change within the rainfall and fertility*
*transect?*
As mentioned above, we have now provided more information as to the relative hierarchy of the
environmental drivers that we monitored. We agree that the abundance/activity of functional
microbial groups would play a role and that such a dataset would definitely provide additional
insight into trace gas fluxes along these gradients. However, as we did not take those
measurements as part of this study, discussing how they may have affected our results would be
purely speculative. We have added a sentence into the discussion to specifically mention that
point (i.e. that in future studies, measurement of functional groups could add additional insight;
Lines 611-612).
*(4) Without any additional literature reference the transfer from Tamai et al., 2003 for*
*methanotrophs to methanogens is hard to buy. In Tamai et al., 2003 a negative correlation*
*between CH4 uptake rate and Al was found. Table 2 shows that your inhibition might be possible*
*for P8, P19, P32, but not for the others. However, these 3 sites show actually the lowest CH4*
*fluxes in the rain season 2011 (Fig. 2). Shouldn't a correlation of net flux and Al result in a*
*positive correlation if inhibition of methanotrophs based on Tamai et al., 2003 is assumed? If*
*your assumption would be valid, how can you explain a simultaneous inhibition of*
*methanotrophs which could cancel out your inhibition of methanogens? Since*
*methanotrophs and methanogens are different functional groups of microbes, I think this is*
*speculative.*
We agree with this comment and this was removed in the revised version of section 4.2.
*(5) For me it seems more plausible that a combination of pH, BS and ECEC which show strong*
*correlations as well, might result a stronger impact for CH4 flux. And a correlation of 15N*
*might point towards coupled methane oxidation and denitrification (e.g. Zhu et al., 2016)?*
As mentioned above, we have substantially altered Section 4.2, which is the section of the
discussion related to $CH_4$ fluxes. However, as we outlined in the comment above, our results do
not show any correlation between $CH_4$ and $N_2O$ fluxes, in the annual or seasonal averages, so we
chose not to incorporate that into the discussion.
*Based on the microbial processes it can be assumed that CH4 oxidation should contribute to*
*CO2 formation. However, this is indicated by a correlation of only -0.24 (CH4 and CO2) in*
*Table 5. Consequently, a potential coupling of aerobic methane oxidation and denitrification*
*might result only -0.07 (CH4 and N2O) in table 5.*
The correlation coefficients referred by reviewer 1 here are not statistically significant and
therefore we chose not to incorporate them into the discussion. Additionally, even granting that
this assumption of CH4 oxidation contributing to CO2 formation is valid, by looking at the
magnitude of soil CO2 fluxes in comparison to soil CH4 uptake (Figs. 2a-b), such a contribution
would be minute compared to the more conventional contributions of heterotrophic (oxidation of organic C with O2) and autotrophic (plant roots) respiration. As to possible coupling of CH4
oxidation with denitrification, please see our answer to the same comment above.
*Finally the introduction and discussion would highly benefit to be focused more on microbial*
*processes.*
We have changed the introduction in response to this comment. Although we still start with a
general intro about trace gases from Central and South American forests, and possible
temporal/spatial controlling factors, we then proceed to introduce each trace gas individually,
before moving on to introduce the gradient study. See Lines 59-106.
Minor comments:
Introduction
*It might be better for the reader to follow the different microbial processes which cause the*
*production and consumption of each trace gas rather than jump from effects of temperature to*
*moisture to soil properties on CO2, CH4, N2O and NO? Overall the introduction is missing a*
*clear structure.*
See comment above.
*You are writing about methanotrophs and methanogens, but for the other trace gases you don't*
*include any information about the processes and functional microbial groups.*
In response to this and the other comments above, we now introduce each trace gas individually,
briefly commenting on the processes of importance (i.e. autotrophic/heterotrophic respiration,
methanotrophs/methanogens, nitrification/denitrification; See Lines 59-106).
*Line 40: Studies (without references) either include references or refer to a comprehensive list in*
*supplement.*
This sentence has been revised so that the vague reference to "studies" is gone. (However,
annual soil trace gas fluxes in Central and South American (CSA) tropical lowland forests can
vary significantly; in one study…; Line 43)
*Line 65/66: take care of terminology, maybe define once? Net CH4 flux consists of production*
*(positive) and consumption (negative). Furthermore, it should be mentioned that production*
*occurs even under negative net CH4 flux, but consumption is predominant.*
This sentence has been revised as follows: Soil $CH_4$ fluxes (predominant flux indicated by
positive values (net emissions) or negative values (net consumption)) in CSA tropical lowland
forests… (Line 79)
Material and Methods
Line 149 "soil trace gas flux measurement": you can only measure mixing ratios. Fluxes are the
result of a second order calculation.

Line 150 "fluxes were measured"?
We have changed the title of section 2.2 to "soil trace gas flux calculation" and altered the
wording in that section to indicate that we determined fluxes rather than directly measuring
them.
*Line 168 Please specify what gas did you flow through the chambers? Ambient air, synthetic air?*
Line 246 now specifies ambient air.
*I recommend including the formulas to calculate CO2, CH4, N2O (static) and NO (dynamic),*
*plus the trapezoid rule to calculate the annual fluxes that the reader does not have to look up*
*several other papers to follow the calculations.*
Fluxes were calculated using the linear change in concentration over time (now included at Lines
197 and 203). The trapezoid rule is an established method of filling in gaps between sample
dates by assuming a linear relationship in gas fluxes between those two dates (Line 207-280).
Neither of these calculations uses a specific formula.
*Results*
*The results are majorly focusing on the descriptive correlations. Why the major results of CO2,*
*CH4, N2O, NO fluxes is not presented here? For me these are the major results obtained from*
*the field by hard work (Fig1 and Fig2).*
The raw data can be made available for teams developing models and/or needing more specific
information, but as the data was presented in Figure 1 and Figure 2, we chose to focus the results
and discussions on patterns that we found in the data.
*Line 291 Due to different soil properties for each site, it seems not very helpful to present Fig. 3*
*and talk about a "parabolic relationship".*
Please see our related comment above. As shown in the figure above, even once the different
sites are identified with unique symbols, the data do not separate out, but instead, together,
exhibit this parabolic relationship. It is also shown in Table S2 that moisture was a major
controlling factor during the dry season and within each individual site.
Discussion
*Statement about what might cause the NO3- differences? Wet deposition, if yes, are there values*
*from literature?*
We have measured the gross rates of soil-N cycling at these sites. The rates of gross N
mineralization (2-5 mg N $kg^{-1}$ $d^{-1}$, or about 68-170 mg N/m2/day in the top 5-cm depth, using
our measured soil bulk density, averaged across sites, of 0.68 g/$cm^3$) and gross nitrification (1.2-
2.4 mg N $kg^{-1}$ $d^{-1}$, or about 41-82 mg N/m2/day in the top 5-cm depth) were much higher than
our measured wet N deposition (9 kg N/ha/yr or only 2.4 mg N/m2/day) at the Gigante site (see
map in Fig. S1; Gigante is across the Panama Canal from our present sites).

We would actually not assume that the mineral N in the soil is directly influenced by the external
N input via wet N deposition. The soil N cycling rates are much larger than the wet deposition,
based on our previous sites in Gigante (e.g. Corre et al. 2010, 2014), Ecuador (Baldos et al.
2015), and Indonesia (Allen et al. 2016).
We discussed the pattern of the soil NO3- levels among sites, or the mineral N pool for that
matter, in perspective of the soil-N cycling, which influencse this mineral N levels, and
ultimately reflected in our overall index of soil N availability status (low or high N availability),
15N natural abundance enrichment factor (which has been shown to correlate with soil N
availability; see lines 228-237 of the manuscript).
Thus, we decided not to include in our discussion about wet deposition, which obviously will not
directly influence the soil $NO_3^-$ levels, but discussed the patterns of $NO_3^-$ among sites with
regards to soil N availability status of the sites.
*The connection of the trace gas fluxes to microbial processes is missing. E.g. the correlation of*
*CH4 fluxes (net uptake) is negatively correlated to 15N natural abundance. Does this point*
*towards a CH4 production coupled to denitrification? And could this coupling be less relevant in*
*the dry season versus the wet season and thereby result amplified correlations in the dry season?*
The negative correlation of CH4 fluxes (net uptake) with 15N natural abundance was indeed
discussed (lines 488-493). However, please refer to our explanation above as to why we don't
think that this relationships points towards CH4 production being coupled to denitrification.
*Figures:*
*Error bars are missing for Fig 3, 4, and 5*
*Fig. 4 a, b, c should include a 0 line for easier understanding. Fig. 4a might be better to bin data*
*into moisture classes of 10%. Less data points will make the figure easier to understand and*
*better show trends. Error bars can be included. Would it make more sense to average the single*
*points and report error bars to highlight the grouping in different fertilizer regimes Fig 4b? That*
*might be helpful for discussion?*
We chose not to put error bars on the scatterplots, as their purpose was to highlight trends, which
may have been masked by including so much additional information. However, in response to
this comment, we have included a zero line for 4a (zero occurred at the top of b and c). We are
reluctant to average the data, however, as the current figures allow readers to see the exact spread
found within each site rather than simply the standard deviation shown on error bars.
*Fig. 5: Where was the NO ambient mixing ratio measured? Close to the ground (chamber*
*height) or 2m height? Are there references available for such high NO ambient mixing ratios*
*and possible sources? Based on Remde et al (1989) it might be helpful to plot NO release rate*
*versus ambient NO mixing ratio at same moisture and temperature for each site. Furthermore,*
*only data points for a range of soil moisture and soil temperature should be selected.*

The NO ambient mixing ratio was measured at a height of 2 m above the ground (prior to each
chamber measurement) near to each of the 4 chamber locations at each of the 4 replicate plots
per site on each sampling day. (added at Line 179-181)
As to the last comments (*to plot NO release rate versus ambient NO mixing ratio at same*
*moisture and temperature for each site; only data points for a range of soil moisture and soil*
*temperature should be selected),* this would not be meaningful for our data sets, because such a
way of analyzing data is driven by an inherent assumption that the ambient NO mixing ratio is
influenced by biological processes in the soil. This is not the case at our study sites where
anthropogenic ambient NO levels are prevalent, especially the site near to the Panama city and
even the other sites along the Panamal canal, brought about by large shipping traffic. Such high
NOx emission was also reported by Hietz et al. 2011.

[revised manuscript text omitted]
., 2013), soil $CH_4$ fluxes are strongly regulated by soil moisture content. Soil $CH_4$ fluxes from our sites exhibited this expected pattern, with regards to less uptake during periods of high water content (i.e. wet vs. dry season; Table 3), soil moisture being the dominant controlling factor at each site and across sites during each season (Table S1), as well as a positive correlation of soil $CH_4$ fluxes with water content (Fig. 4a). This The dominant role of soil moisture can we attributed to limited gas diffusivity from the atmosphere into the soil and/or methanogenic activity during periods of high moisture. Another soil factor controlling the temporal soil $CH_4$ uptake in our sites may have been soil $NO_3^-$, as we observed increased $CH_4$ uptake as $NO_3^-$ concentrations increased in P8, P19 and

P32 (see 3.3) and it was a dominant controlling factor across sites in both seasons (Table S1).

Although this could be a co-correlation between soil $NO_3^-$ concentration and soil moisture (see 3.1), increasing $CH_4$ uptake in the soil with increasing mineral N has been observed in tropical forest soils of Australia (Kiese et al., 2003), Panama (Veldkamp et al., 2013)

and Indonesia (Hassler et al., 2013). Additionally, our soils exhibited a correlation between annual soil $CH_4$ fluxes and soil $^{15}N$ natural abundance signatures (Table 5), the latter being an indicator of soil N availability (Sotta et al. 2008; Arnold et al. 2009; Baldos et al. 2015). When separated by season, the correlation between soil $CH_4$ fluxes and soil $^{15}N$ natural abundance was stronger in the dry season than the wet season (Table S21), supporting our claim that soil N

availability enhanced $CH_4$ uptake in soils when gas diffusion was favorable (dry season).

The control of soil fertility on the long-term pattern of soil $CH_4$ fluxes across sites was depicted by a correlation between annual soil $CH_4$ fluxes and our calculated soil fertility index (Fig. 4b), which exhibited an opposite pattern to that of annual precipitation (Figure S2).

This soil fertility control was supported by the strong correlations of both annual (Table 5) and seasonal (Table S2) soil $CH_4$ fluxes with ECEC and exchangeable Al, both included in the soil fertility index (Figure S2; see 2.4). The correlations between soil $CH_4$ fluxes and fertility indicators reflected the site differences in soil biochemical characteristics (Table 2). Specifically, as shown by the strong inverse correlation between soil $\delta^{15}N$ natural abundance signatures and exchangeable cations (Table 5), the positive correlation between soil $CH_4$ flux and fertility (Fig.

4b) likely reflected the long-term effects of soil development (Tables 1 and 2) - more $CH_4$ uptake occurred in highly weathered soils with less rock-derived nutrients but high soil N availability (i.e. high $\delta^{15}N$ natural abundance signatures) (Tables 4 and 5). This supports our hypothesis that soil $CH_4$ uptake reflected the control of soil moisture and N availability across sites along this precipitation gradient. Our results also highlight the importance of considering soil properties - in particular the degree of soil development - rather than simply climatic factors, when predicting/modeling soil $CH_4$ fluxes on a large scale.

The negative correlation between annual soil $CH_4$ uptake and annual precipitation (Fig.

4c; see 3.3) seemed at first to conflict with the mechanism we explained above for the positive correlation with soil moisture content (Fig. 4a). However, we attribute this to the fact that annual precipitation was not the underlying factor controlling the annual soil $CH_4$ fluxes across these sites. Instead, the best indicator for annual soil $CH_4$ flux across the five sites was soil fertility (Fig. 4b), which showed an opposite pattern to that of annual precipitation (Figure S2). This soil fertility control was supported by the strong correlations of both annual (Table 5) and seasonal (Table S21) soil $CH_4$ fluxes with ECEC and exchangeable Al, both included in the soil fertility index (Figure S2; see 2.4). The negative correlation of soil $CH_4$ fluxes with exchangeable Al, which was clearly observed in the wet season (Table S21), could suggest an inhibition of methanogens by water-soluble Al (as opposed to inhibiting methanotrophs, as seen by Tamai et al., 2003). 
[revised manuscript text omitted]

---

## Editor Decision (ED1)

Thus, the NO uptake that we  observed may have been driven by both chemical (Pape et al. 2009) and microbiological  processes (as NO is an intermediate product of nitrification and denitrification; Davidson et al. 2000). The dominance of a chemical reaction of NO uptake at our sites was supported by the fact that we observed a negative correlation of soil NO fluxes with ambient air NO concentrations (i.e. net NO uptake increased as ambient air NO concentration increased; Fig. 5). The reaction time of NO with $O_3$, which is then subsequently removed from the enclosed chamber air and deposited onto the soil, is  controlled by the ambient air NO concentrations (Pape et al. 2009). This can occur in under a minute (which we observed on days with low ambient air NO concentrations when we measured net soil NO emissions; e.g. at P8 during the dry season, Fig. 2b) or can take up to the same order of magnitude as the turnover time of the chamber air (which we observed on days with high ambient air NO concentrations when we measured net NO uptake; e.g. at the Met site on most of the sampling days, Fig. 2b). It is notable, that an earlier study in Gigante, which is also part of the Panama Canal watershed, did not show net NO uptake but instead small net NO emissions (Koehler et al., 2009b; Corre et al. 2014). However, as mentioned above, the Gigante site had higher soil N-cycling rates (Corre et al. 2010) and lower ambient air NO concentrations than our sites, such that NO production in the soil overrides the chemical reaction of NO uptake and thus resulted in net soil NO emissions.

**Comment [IT1]:** I have some doubt about that. This may also indicate soil uptake of NO.

**Comment [IT2]:** How did you determine this without any O3 measurements? How long is the turnover time of the chamber air?

As long as the chemical reaction is faster than the residence time in the chamber, there will be significant removal of NO by reaction with O3.

**Comment [IT3]:** I do not agree that the chemical reaction should be considered as part of the NO flux. To my opinion it is a measurement artefact that should be avoided.

---

## Author Response (AR2)

Answer to Reviewer #2:

There is no doubt, that the rainfall shows a "parabolic pattern", however, after showing the individual datapoints in fig.3 the former point cloud now shows linear declines of CO2 flux for each site. Therefore, I assume that the autors mean by "parabolic pattern" a maximum CO2 flux which is limited by a parabolic function? If that is the case, I recommend that the authors include a parabolic function into Fig. 3 and better describe what "parabolic pattern" actually means to them. Otherwise, I recommend to discuss the linear decline with existing literature, Skopp et al. 1990.

In response to this comment, we have now shown the parabolic curve on Figure 3 and included the equation, R-squared and P-value in the figure heading (Lines 38-40). Although it may not be clear on the scatterplot, if we showed linear regressions for the individual sites, these would in fact show an increase of CO2 with soil moisture. However, remember that we are using these sites as a surrogate variable to represent a range of soil conditions, and as such we want to look at the relationship across sites; this relationship was parabolic, as we now show in Figure 3 and have observed previously in Panama (Koehler et al. 2009, fig. 3), in Indonesia (van Straaten et al. 2011, fig. 5), in Costa Rica (Schwendenmann et al. 2003, fig. 3), and in Brazil (Sotta et al. 2006, fig.4).

My intention was not to motivate the authors for including a sentence about using microbial abundance and activity in future studies, but rather to discuss their findings in a better context to microbial processes. As I learned from the replies to R1, there is another paper on its way which might deal with that topic? However, also this paper would highly benefit if the coupling of CH4 and N2O fluxes would be discussed. I agree that denitrification is not making sense. However, especially in the dry season when CH4 uptake is dominant and soils are more oxic, also denitrification should be decreased. Since methanotrophs are also known to produce N2O, the decrease of N2O caused by denitrification might be larger than the increase of N2O caused by methanotrophy and therefore no clear correlation between CH4 and N2O is obtained.

Yes, there is an N-cycling paper planned. For this paper, we agree that a discussion of the coupling of CH4 and N2O fluxes would have been really interesting if our results had supported such a discussion. However, there was simply no evidence of this relationship based on our correlation analysis, quite possibly for the reasons suggested in this comment. To incorporate that idea into our discussion, we have now expanded the first paragraph of Section 4.5 (Lines 606-612) to note that we may have missed correlations between gross production/consumption processes belowground, as we were only measuring net fluxes at the soil surface.

I agree that the ambient NO mixing ratio is not the right one to use in a scatter plot, however, I would use the NO mixing ratio shortly before reopening the chamber. Also I think it might be worth to include a statement about potential effects of a variable NO background (before closing and after opening the chamber) on the NO flux measured in between.

We disagree that it would be better to use the NO mixing ratio inside the chamber, particularly during the period shortly before reopening it, as the NO concentration at that time is already the net effect of the chemical reaction (deposition onto the soil within the chamber through reaction of ambient NO with ambient $O_3$; Pape et al. 2009) and microbiological processes (NO consumption
in the soil as an intermediate product of nitrification and denitrification; Davidson et al. 2000).
That would not support our discussion of how the net NO flux was driven by the ambient NO
concentration as well as the NO production/consumption capacity of the soils across those periods
of measurements. Regarding the variable NO background, this is shown in Figure 5 (i.e. the
relationship between ambient NO concentration and soil NO flux) and is discussed extensively in
the revised version of Section 4.4, with the dialogue concerning the chemical
reactions/microbiological processes resulting in net negative NO fluxes (see Lines 565-580 in
Section 4.4 of the manuscript).
Answer to the Subject Editor:

Thus, the NO uptake that we  observed may have been driven by both chemical (Pape et al. 2009) and microbiological  processes (as NO is an intermediate product of nitrification and denitrification; Davidson et al. 2000). The dominance of a chemical reaction of NO uptake at our sites was supported by the fact that we observed a negative correlation of soil NO fluxes with ambient air NO concentrations (i.e. net NO uptake increased as ambient air NO concentration increased; Fig. 5). The reaction time of NO with $O_3$, which is then subsequently removed from the enclosed chamber air and deposited onto the soil, is  controlled by the ambient air NO concentrations (Pape et al. 2009). This can occur in under a minute (which we observed on days with low ambient air NO concentrations when we measured net soil NO emissions; e.g. at P8 during the dry season, Fig. 2b) or can take up to the same order of magnitude as the turnover time of the chamber air (which we observed on days with high ambient air NO concentrations when we measured net NO uptake; e.g. at the Met site on most of the sampling days, Fig. 2b). It is notable, that an earlier study in Gigante, which is also part of the Panama Canal watershed, did not show net NO uptake but instead small net NO emissions (Koehler et al., 2009b; Corre et al. 2014). However, as mentioned above, the Gigante site had higher soil N-cycling rates (Corre et al. 2010) and lower ambient air NO concentrations than our sites, such that NO production in the soil overrides the chemical reaction of NO uptake and thus resulted in net soil NO emissions.

**Comment [IT1]:** I have some doubt about that. This may also indicate soil uptake of NO.

**Comment [IT2]:** How did you determine this without any O3 measurements? How long is the turnover time of the chamber air?

As long as the chemical reaction is faster than the residence time in the chamber, there will be significant removal of NO by reaction with O3.

**Comment [IT3]:** I do not agree that the chemical reaction should be considered as part of the NO flux. To my opinion it is a measurement artefact that should be avoided.

Comment 1: We have incorporated this comment into the revised last sentence of Lines 565-580
by not claiming the 'dominance' of chemical reaction of NO over that of microbially-mediated
NO consumption in the soil.
Comment 2: We decided to delete this sentence in the revised version as this is actually not
necessary to support our argument. However, in answer to this question: we calculated the
turnover time of the enclosed chamber air by chamber vol. (11 L) ÷ sampled air flow rate (0.5-
0.6 L/min) (Lines 178 & 182). This statement was based on our results which we included during
the initial review (see below Fig. 1a-b).

Comment 3: We agree that NO 'uptake' may not be the best term to use, so in the Results section
we include both terms (previous line 388 now reads 'In all five sites, net NO uptake or negative
NO flux ….') and then throughout the manuscript we now use 'net negative NO flux'. However,
we disagree that the reaction of ambient NO with ambient $O_3$ within the chamber prior to the
measurement system (where the sampled gas is passing through the $CrO_3$ – luminol – detector)
should be termed a measurement artifact. This principle of NO measurement by Scintrex LMA-3
chemiluminescence is an established method for field studies in the tropics that have used the
dynamic chamber method (Veldkamp et al. 1998, Verchot et al. 1999, Hall and Matson 2003,
Keller et al. 2005, Purbopuspito et al. 2006, Koehler et al. 2009; Hassler et al. 2017). The
reaction of  NO with ambient $O_3$ normally happens within a few seconds after chamber closure
(see Fig 1a), and hence is usually overshadowed by the linear change of NO concentrations
during the 5- to 7-minute measurement of chamber closure. We observed such long reactions of
ambient NO with O3 within the chamber (see Fig. 1b) during periods and/or in sites that had
high O3 concentrations due to the proximity to O3 sources. Thus, it is not the measurement
method but the unusually high O3 concentrations that led to net negative NO fluxes. Therefore,
instead of terming these negative NO values as measurement artifacts, we include in our revised
discussion that these negative NO fluxes are caused by both chemical and microbiological
reactions.

Other changes made in this version:
Line 10: contact email for A. Matson was updated
Line 66/439: spelling mistake in author name corrected
Line 160/177: terminology standardized for chamber methods (using 'vented' in both cases)
Table 4: changed formatting to match the other tables

[Figure]

Fig. 1. Typical measurements depicting net
NO emission (usually occurred when
ambient NO concentration was low) and
net negative NO flux (usually occurred
when ambient NO concentration was high).

[revised manuscript text omitted]

NO  fluxes that we  observed may have been driven by both chemical reactions (deposition onto the soil within the chamber through reaction of ambient NO with ambient $O_3$;

Pape et al. 2009) and microbiological  processes ( NO consumption in the soil as  an intermediate product of nitrification and denitrification; Davidson et al. 2000).

The reaction time of NO with $O_3$, which is then subsequently removed from the enclosed chamber air and deposited onto the soil, is controlled by the ambient air NO concentrations (Pape et al. 2009).

It is notable, that an earlier study in Gigante, which is also part of the

Panama Canal watershed, did not show  negative NO  fluxes but instead small net NO

emissions (Koehler et al., 2009b; Corre et al. 2014). However, as mentioned above, the Gigante site had higher soil N-cycling rates (Corre et al. 2010) and lower ambient air NO concentrations than our sites, such that NO production in the soil  may have compensated the chemical reaction of ambient NO  with $O_3$ and thus resulted in net soil NO emissions. Contrary to this, the negative correlation of soil NO fluxes with ambient NO concentrations observed in our sites (i.e. net negative NO flux increased as ambient air NO concentration increased; Fig. 5)

suggests that NO production in the soil was overshadowed by the chemical reaction of ambient

NO with $O_3$ and thus resulted in net negative NO fluxes.

The general trend across sites did not support our hypothesis regarding soil NO emission, since local conditions of high ambient NO concentrations in the atmosphere had an overriding effect resulting in net NO uptake in soils (Fig. 2d). However, our results indicated that our soils could also be a net source of NO when soil conditions were favourable and/or ambient air NO

concentrations were not elevated. We observed that net NO uptake was consistently higher in the wet season than the dry season (Table 3); in the dry season, when aerobic soil conditions prevailed due to low soil moisture contents (Table 3), NO production in the soil may have been more favoured (Conrad, 2002), partly counteracting the chemical reaction of NO removal from the atmosphere and its deposition onto the soil. This is also supported by the negative correlation between dry-season soil NO fluxes and clay contents of the sites (Table S2), suggesting that soil

NO fluxes were responding to conditions favourable for NO production. Favourable soil conditions were most visible at P8, which had the highest soil NO emissions (with low ambient air NO concentrations) in the dry season (Table 3; Fig. 2d); soil NO fluxes at this site increased when aerobic soil conditions prevailed (i.e. negative correlation with soil moisture; see 3.5) and increased with substrate availability (i.e. positive correlation with soil $NO_3^-$; see 3.5).

In summary, although the soils in our study sites can be a net source of NO, particularly during the dry season (Fig. 2d) and in sites where ambient air NO concentrations were low (Fig.

5), most of the time the soils acted as net sink of NO, signifying the importance of soil and vegetation as NO sinks (Jacob and Bakwin, 1991; Sparks et al., 2001) in areas affected by anthropogenic NO sources.

**4.5 Implications for climate change**

It is notable that, although all four trace gases were strongly correlated with the temporal variation in soil moisture and had clear differences between seasons (Table 3), there were no correlations between the four soil trace gases when looking at the annual fluxes (Table 5) or seasonal averages (Table S2). This lack of correlation may be due to the interaction of other soil/climatic factors with known drivers of soil trace gas production and consumption. It may also reflect that trace gas fluxes at the soil surface are the net result of gross production/consumption processes occurring belowground, where correlations may exist

[revised manuscript text omitted]
_2$ (Mg C ha$^{-1}$ yr$^{-1}$) | $CH_4$ (kg C ha$^{-1}$ yr$^{-1}$) | $N_2O$ (kg N ha$^{-1}$ yr$^{-1}$) | NO (kg N ha$^{-1}$ yr$^{-1}$) |
|---|---|---|---|---|
| Met (1700 mm) | 8.48 (0.70) | -0.34 (0.17) | 0.41 (0.06) | -0.82 (0.16) |
| P27 (2030 mm) | 9.16 (0.62) | -0.51 (0.04) | 0.43 (0.06) | -0.12 (0.04) |
| P8 (2360 mm) | 10.14 (0.76) | -1.45 (0.15) | 1.07 (0.15) | -0.17 (0.17) |
| P19 (2690 mm) | 9.89 (0.49) | -1.98 (0.07) | 0.35 (0.05) | -0.21 (0.10) |
| P32 (3400 mm) | 7.89 (0.84) | -0.94 (0.19) | 0.66 (0.18) | -0.03 (0.09) |

[a] Calculated using the trapezoidal rule between fluxes and time interval, covering the measurement periods of January - December 2011 for $CO_2$ , $CH_4$ and $N_2O$, and June 2010 - May 2011 for NO.

Annual fluxes were not tested statically for differences among sites since these are trapezoidal extrapolations.

Table 5 Spearman correlations of soil biochemical characteristics[a] and annual (measured in 2011) soil trace gas fluxes from five lowland tropical forests along orthogonal precipitation and fertility gradients in the Panama Canal watershed, central Panama.

| | ECEC | BS | Na | Al | pH | Clay | $CO_2$ | $CH_4$ | $N_2O$ | NO |
|---|---|---|---|---|---|---|---|---|---|---|
| $^{15}N$ sig. | -0.87** | -0.67** | -0.30 | 0.42 | -0.61** | -0.15 | 0.41 | -0.70** | 0.30 | 0.16 |
| ECEC | | 0.80** | 0.34 | -0.50 | 0.76** | -0.12 | -0.33 | 0.77** | -0.09 | -0.17 |
| BS | | | -0.13 | -0.87** | 0.96** | -0.12 | -0.40 | 0.78** | -0.12 | -0.54 |
| Na | | | | 0.45 | -0.18 | -0.15 | 0.04 | 0.01 | -0.01 | 0.60** |
| Al | | | | | -0.87** | 0.04 | 0.24 | -0.71** | 0.17 | 0.58** |
| pH | | | | | | -0.04 | -0.34 | 0.76** | -0.12 | -0.54 |
| Clay | | | | | | | -0.13 | -0.17 | -0.67** | -0.34 |
| $CO_2$ | | | | | | | | -0.24 | 0.26 | 0.10 |
| $CH_4$ | | | | | | | | | -0.07 | -0.31 |
| $N_2O$ | | | | | | | | | | 0.19 |

** $P < 0.01$, $n = 20$ (4 replicate plots in each of the 5 forest sites)

[a] Soil parameter abbreviations: $^{15}N$ natural abundance signature ($^{15}N$ sig.), effective cation exchange capacity (ECEC) and base saturation (BS).

[Figure]

**Fig. 1** Mean (±SE, n = 4) soil **(a)** temperature, **(b)** moisture, **(c)** $NH_4^+$ and **(d)** $NO_3^-$ concentrations measured in the top 5 cm of soil in lowland forests along orthogonal gradients of annual precipitation and soil fertility in the Panama Canal watershed, central Panama.

Gray shading indicates the dry season (January through April).

[Figure]

**Fig. 2** Mean (±SE, n = 4) soil **(a)** $CO_2$, **(b)** $CH_4$, **(c)** $N_2O$ and **(d)** NO fluxes from lowland forests along orthogonal gradients of annual precipitation and soil fertility in the Panama Canal watershed, central Panama. Gray shading indicates the dry season (January through April).

[Figure]

Fig. 3 Soil $CO_2$ fluxes and moisture contents (top 5 cm) in five lowland forests along orthogonal gradients of annual precipitation (shown in brackets) and soil fertility in the Panama Canal watershed, central Panama. Each data point is the average of four replicate plots on one sampling day from one of the five sites, measured from June 2010 to February 2012 (n = 145); the quadratic regression across sites (shown) is: $y = -321.1x^2 + 517.8x - 81.2$ ($R^2 = 0.30$, n = 145, P<0.01).

[Figure]

[Figure]

[Figure]

Fig. 4 Average daily soil $CH_4$ fluxes plotted against (a) soil moisture (top 5 cm), and annual soil $CH_4$

fluxes plotted against (b) soil fertility index and (c) annual precipitation. For (a), each data point is the average of four replicate plots on each sampling day of each of the five sites, measured from June 2010

to February 2012. The five lowland forests are located along orthogonal gradients of annual precipitation and soil fertility in the Panama Canal watershed, central Panama.

[Figure]

**Fig. 5** Soil NO fluxes plotted against ambient air NO concentrations; each data point is the average of four replicate plots on each sampling day in each of the five sites, measured from June 2010 to June

2011. The five lowland forests are located along orthogonal gradients of annual precipitation and soil fertility in the Panama Canal watershed, central Panama.